# Effective DDoS attack detection in software-defined vehicular networks using statistical flow analysis and machine learning

**Himanshi Babbar**[1], **Shalli Rani**[1]*, **Maha Driss**[2,3]

**1** Chitkara University Institute of Engineering and Technology, Chitkara University, Punjab, Rajpura, India, **2** RIOTU Lab, CCIS, Prince Sultan University, Riyadh, Saudi Arabia, **3** RIADI Laboratory, National School of Computer Sciences, University of Manouba, Manouba, Tunisia

These authors contributed equally to this work.

* shallir79@gmail.com

## Abstract

Vehicular Networks (VN) utilizing Software Defined Networking (SDN) have garnered significant attention recently, paralleling the advancements in wireless networks. VN are deployed to optimize traffic flow, enhance the driving experience, and ensure road safety. However, VN are vulnerable to Distributed Denial of Service (DDoS) attacks, posing severe threats in the contemporary Internet landscape. With the surge in Internet traffic, this study proposes novel methodologies for effectively detecting DDoS attacks within Software-Defined Vehicular Networks (SDVN), wherein attackers commandeer compromised nodes to monopolize network resources, disrupting communication among vehicles and between vehicles and infrastructure. The proposed methodology aims to: (i) analyze statistical flow and compute entropy, and (ii) implement Machine Learning (ML) algorithms within SDN Intrusion Detection Systems for Internet of Things (IoT) environments. Additionally, the approach distinguishes between reconnaissance, Denial of Service (DoS), and DDoS traffic by addressing the challenges of imbalanced and overfitting dataset traces. One of the significant challenges in this integration is managing the computational load and ensuring real-time performance. The ML models, especially complex ones like Random Forest, require substantial processing power, which necessitates efficient data handling and possibly leveraging edge computing resources to reduce latency. Ensuring scalability and maintaining high detection accuracy as network traffic grows and evolves is another critical challenge. By leveraging a minimal subset of features from a given dataset, a comparative study is conducted to determine the optimal sample size for maximizing model accuracy. Further, the study evaluates the impact of various dataset attributes on performance thresholds. The *K*-nearest Neighbor, Random Forest, and Logistic Regression supervised ML classifiers are assessed using the BoT-IoT dataset. The results indicate that the Random Forest classifier achieves superior performance metrics, with Precision, F1-score, Accuracy, and Recall rates of 92%, 92%, 91%, and 90%, respectively, over five iterations.

**Data Availability Statement:** Data is publicaly avaialble on https://www.kaggle.com/datasets/vigneshvenkateswaran/bot-iot.

**Funding:** The authors would like to acknowledge the support of Prince Sultan University for paying the Article Processing Charges (APC) of this publication. The funders had no role in study design, data collection and analysis, decision to publish, or preparation of the manuscript but they supported in supervision.

**Competing interests:** The authors declare no conflict of interest.

# Introduction and background

VN [1] are proposed to address essential needs of increasing transportation effectiveness and safety, lowering accident rates, and minimizing the effects of severe traffic congestion. Surveillance provisions, traffic control, and mobile vehicular cloud services on VN are no longer a distant possibility. Data is stored and sent between vehicles in VN [2, 3] because there is no centralized control nodes there. This implies that each vehicle node plays a crucial role, comparable to the storage and transmitting duties performed by routers and switches in conventional networks. Vehicle nodes work together to exchange messages and forward data [4]. DDoS attacks significantly lower the amount of data transmission integration among vehicular nodes, minimize the transmission speed of packets and bandwidth of VN, and limit the effectiveness of communication networks. Moreover, DDoS attacks can also have serious repercussions since compromised nodes utilize the resources of networks maliciously and degrade their effectiveness [5].

## Background

The development of VN has a lot of promise thanks to SDN. Owing to the centralized intelligent control that SDN brings, SDVN have many administrative advantages compared to conventional VN [6]. SDN architecture provides the application and control plane to VN, as seen in Fig 1. A selection of services and applications are offered through the applications plane. The controller of the software platform, which serves as the central decision-making hub for an entire SDVN [7] architecture, is accommodated in the control plane, specifically, the control plane of SDN [8]. The control plane also helps aggregate underlying resources and make networks more programmable. Underlying network resources are mostly included in the forwarding plane of SDVN [9]. Hardware for forwarding, known as the forwarding plane, making use of switches and routers with SDN capabilities, is found on the higher data plane. Construction and networked VN, such as Vehicle-to-Infrastructure (V2I) and Vehicle-to-Vehicle (V2V), make up the majority of the lower forwarding plane. Additionally, the Northbound Interface (NBI) controls the opening of the Application Programming Interface (API) for communication between application and control planes [10].

## Network design

SDN has become a technology for interconnected network architecture recently. There are three planes: application, control and forwarding. The control and forwarding planes are separated in SDN, allowing for flexibility and simplification. The SDN controller, located in the control plane, is responsible for managing the entire network. By providing a comprehensive picture of the networks and central control functions, SDN makes it easier to compile network statistics and offers greater network security than conventional methods. The most important protocol in SDN architectures is the south-bound protocol, which facilitates communication between network components and the controller.

We have organized our novel proposed model in the control plane as follows to increase its effectiveness:

- The SDN control plane is completely programmable and customizable.

- The forwarding plane of the control plane can support several networks.

- The model is suitable for IoT networks since it can use IoT devices without burdening them computationally or otherwise.

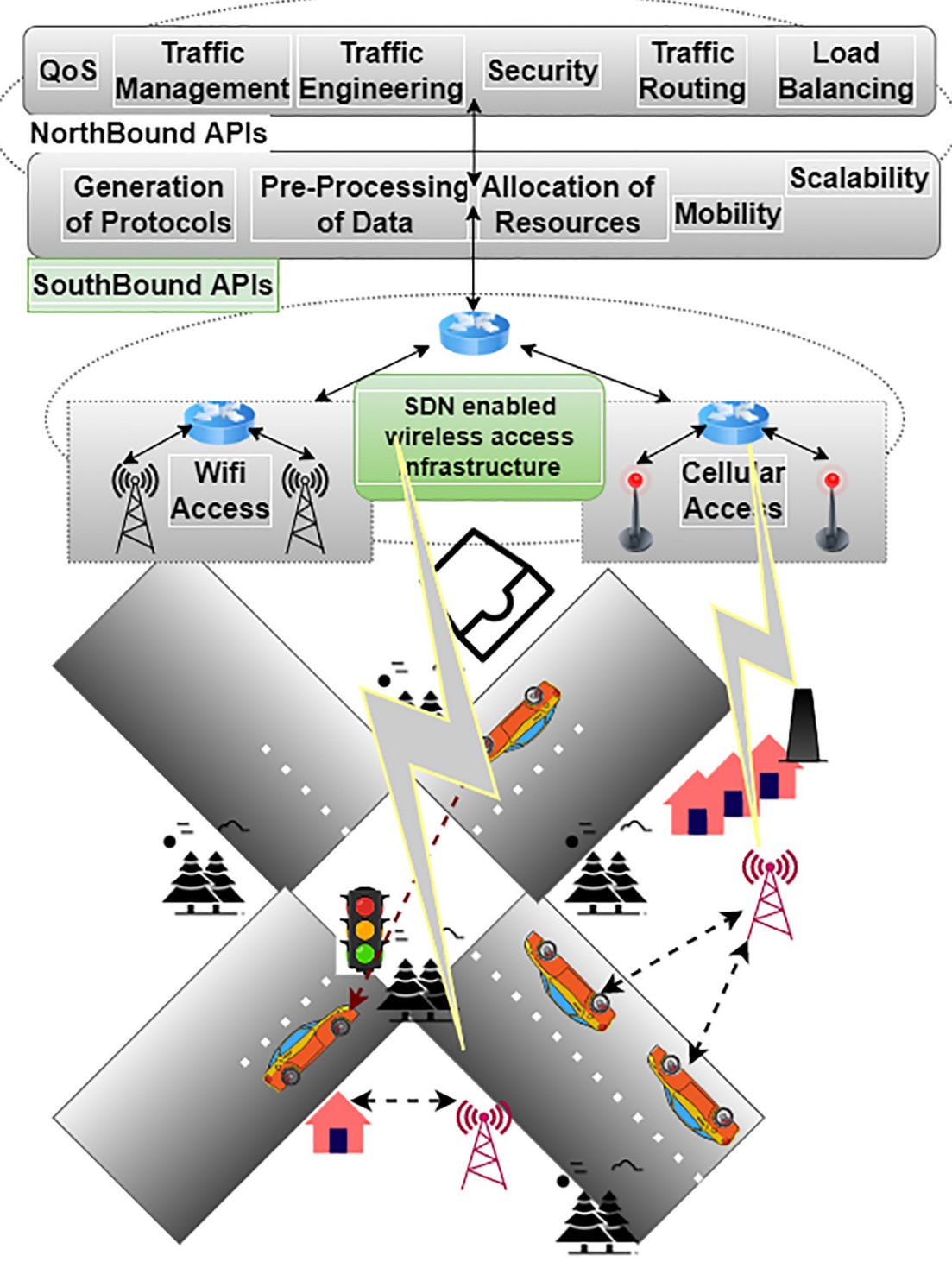

**Fig 1. Software defined vehicular networks.**

- Open-Flow (OF) switches are implemented to address diversity between IoT devices and SDN controllers.

The control and forwarding plane of an SDN architecture identifies OF as a prime southbound protocol. The switches and controllers are related because they create activities and flow tables that instruct the switch on how to handle these channels and flows. A proper method for inspecting network data to find risks, suspicious activity, and attacks is made possible by combining IoT and SDN. In addition, numerous IoT devices, such as sensors, wireless technologies, and smart devices, can be connected to the forwarding plane of SDN. Additionally, NBI indicates the availability of the Application Programming Interface (API) for communication between the application and control plane. The southbound interface is harnessed to allow the API to connect the control plane to the forwarding plane. In the context of networking, controllers communicate back and forth using the API's eastbound and westbound communication capabilities.

## Motivation

Due to cognitive design, Deep Learning (DL) Intrusion Detection Systems (IDS) are excellent at prediction and classification [11–13]. For SDN-IoT [14, 15] networks, DL and ML will produce encouraging outcomes. An SDN-IoT system generates an enormous amount of data, which may be exploited by learning techniques to make better and more informed decisions [16]. Fig 2 showcases the rise of DDoS attacks on numerous applications of IoT based on ML approaches. Security can also be improved by adding intelligence through learning-based methods. The main goal of this research is to develop an IDS for DDoS [17] attacks using DL approaches due to the increased use of various ML/DL models to combat such security and privacy issues. Fig 2 shows that DDoS attacks still happen despite intensive research being done to protect SDN–IoT infrastructure (https://trends.google.com/trends/).

Taking into account some of the issues discussed this far, this research suggests a revolutionary ML-based IDS named SDN-IoT-based ML to foresee different DDoS attack categories [18]. The suggested method has a high degree of accuracy in spotting DDoS attacks. The dataset that was taken into consideration for this research is contemporary and widely deployed in the research community to develop intrusion detection algorithms for SDN-IoT networks. The study also surveys several ML-based models by classifying them as benign or malicious [19]. Implementation of the developed SDN-IoT ML-based IDS is evaluated with current baseline methods, specifically ML-based IDS models.

## Problem statement

The emerging technology SDN is programmable, scalable, and flexible. Furthermore, in IDS, there is an inherent need to develop strategies for reducing the impact of bias in datasets while considering accuracy and simultaneously making use of entropy. This research work proposes the detection of DDoS attacks. First, destination entropy and measurements for flow statistics to differentiate normal and malicious attacks are used. Second, the approach deploys ML algorithms to classify packets as normal or as a malicious attack. In this paper, if the value of entropy is less than the threshold value for a maximum of ten consecutive times, it is regarded as an attack. To cope with the accurate detection of attacks, the optimum value of the threshold must be selected after many different tests on the topologies proposed. To identify an optimal threshold value, a number of tests have been executed to determine how attacks have affected entropy for different types of topologies having different rates of attack.

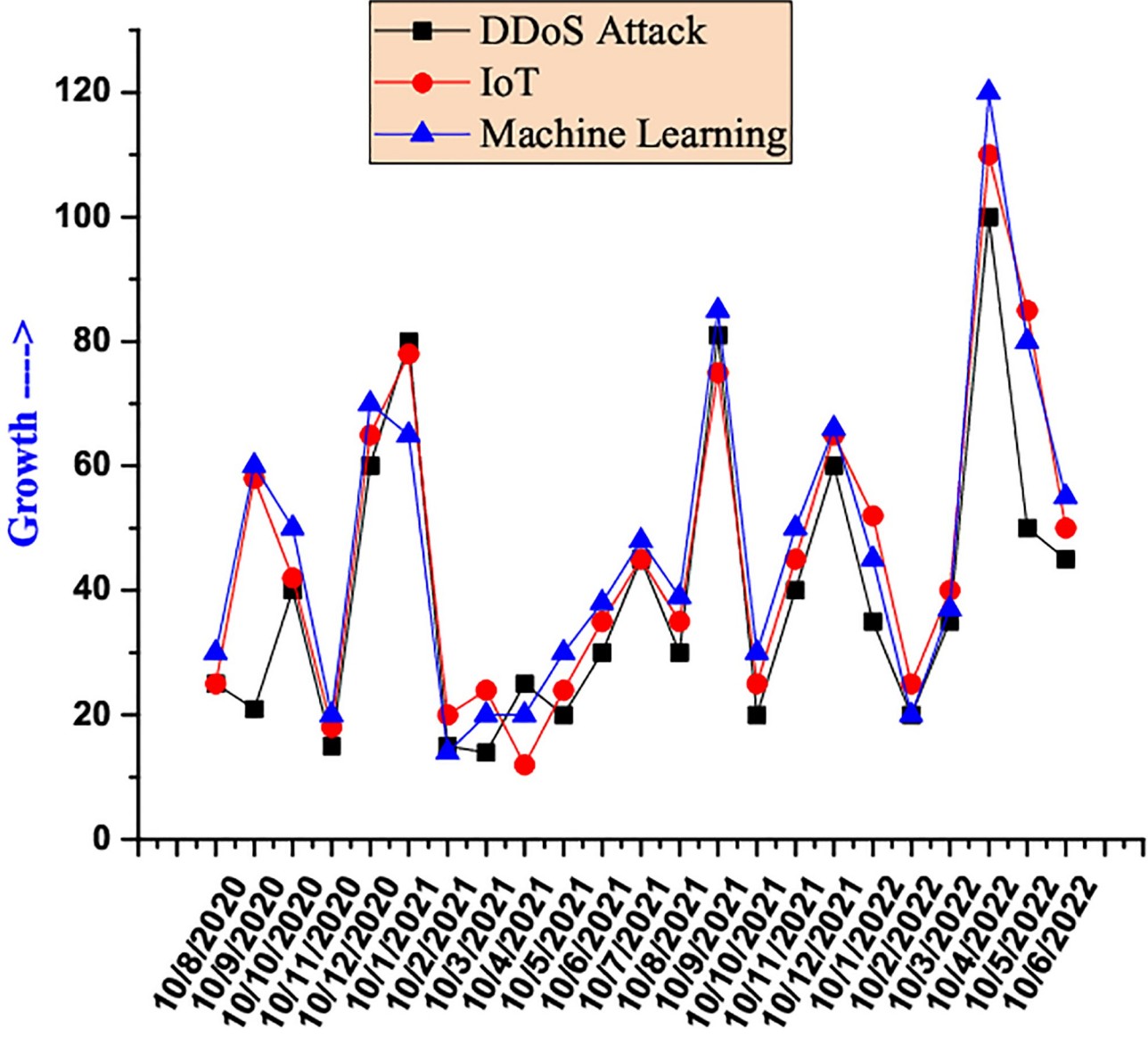

**Fig 2. Development of DDoS attacks, IoT applications and ML techniques.**

## Contributions

The main contributions are:

1. Facilitate comprehensive insight into previous work, focusing on ML techniques that have employed the BoT-IoT dataset for DDoS development.

2. The framework for the proposed work has been showcased where attack detection, mitigation, and recognition are considered. Concerning these considerations, two unique algorithms are proposed for attack detection and traffic categorization.

3. Determine DDoS attack detection effectiveness for a given SDN controller. In this manner, two features (flow length and flow duration) and two approaches are presented, namely

statistics of flow and entropy measurement. To identify DDoS attacks, a model using the degree of attack is presented.

4. Represent detailed data-pre-processing steps utilizing feature extraction and selection techniques.

5. Utilize ML-based models for detecting DDoS attacks on SDN-based IoT employing *K*-Nearest Neighbour (KNN), Logistic Regression (LR) and Random Forest (RF).

6. Our results demonstrate the classification of benign, reconnaissance, DoS, and DDoS traffic with an accuracy of 91% and represent extracted features of a dataset.

## Organization of paper

The paper is organized as follows: Section 2 explains the related work done by the existing authors for the detection of DDoS attacks; Section 3 describes the methodology that shows the impact of the behavior post traffic and identification of DDoS attacks; Section 4 describes the proposed approach for the detection of attacks; Section 5 highlights the experimental setup along with the dataset and results; Finally Section 6 concludes the paper.

## Related work

For the detection of DDoS, ML techniques are considered effective at identifying attacks happening in the control plane of the SDN framework [20]. In this section, previous research deployed on SDN is described.

VN arise from IoT, which is used as a combined network that executes traffic management and intelligent control of traffic in SDN. Therefore, Yu et al. [21] develop a mechanism for detection and quick response to attacks happening in VN based on SDN. The developed mechanism is based on multi-dimensional information and the strategy itself is based on the flow of extracted features rather than the triggering of messages in the OpenFlow protocol. In this work, the results verify that the scheme based on attack detection minimizes starting time and false alarm rate.

Muthanna et al. [22] present a smart, SDN-enabled architecture for effective detection of attacks in IoT deploying Long Short Term Memory (LSTM) that accesses a baseline IoT dataset (CICIDS2017) utilizing evaluation metrics. The developed model acquires 99.50% accuracy for detection having a low false positive rate (FPR). The results are compared with different models in terms of efficiency, precision and various other evaluation metrics. Wani et al. [23] deploy features of SDN that are used to mitigate attacks in DDoS for IoT networks. The method is used to detect abnormal behavior and predict abnormality affected by DDoS attacks using ML [24]. The results show that the precision of this work is 98.74% which is much less than our proposed work. The authors also develop a mechanism of fusion entropy which identifies attacks by computing the randomness of events happening in a network. Simulation results of the developed mechanism show the value of entropy for attack detection that happened to be 99.25% lesser than a normal attack. The key advantage is that their methodology identifies attacks happening at the beginning of an attack which deploys integration between two entropy's that significantly minimizes the entropy value [25]. Sahoo et al. [26] detect attack traffic by undertaking centralized control, therefore, by applying ML, malicious traffic can be detected and a Support Vector Machine (SVM) model is used for minimizing noise caused by differences in features that achieve classification accurately with better generalizations. Sultan

et al. [27] recommended detection and mitigation of DDoS for systems in SDN in which the model is helpful in identifying attack traffic in multi-controller environments.

## DDoS attacks in SDN for vehicular networks

IoT [28], an incorporated network that enforces intelligent management of traffic [29], intelligently changes information services, and vehicle intelligent control [30] in conformity with established communication protocols and data communication standards is where the concept of VN originated. Sedjelmaci et al. [31] propose safeguards against three types of attacks, namely DoS, Integrity Target, and False Alert Generation can be guarded against using effective and lightweight intrusion detection mechanisms for vehicle networks (ELIDV). ELIDV is built on a set of principles that quickly and accurately identify harmful vehicles.

To enable safe communication, it is crucial to ensure security in SDN. Eliyan et al. [32] focus on DDoS attacks in the control plane. A DDoS assault prevents users from accessing a system or network resources. This is accomplished by using all of a network's bandwidth or all network nodes' resources (such as memory and CPU). The various categories of DDoS attacks [33] are:

1. **UDP Flood**: an attack that attempts to bring down the server by flooding the targeted host with numerous UDP packets to various random ports. Attackers typically use the connectionless capability of UDP to send a stream of UDP data packets to target workstations. The target machine's queue fills up and it is unable to react to requests from reputable users [34]. To conceal the positions of attacked machines, the attacker typically hijacks source IP addresses of UDP packets.

2. **SYN Flood**: an attack that targets a victim's computer by initiating a TCP connection. The victim receives a huge amount of SYN packets. However, no ACK is ever returned, allowing the victim's system to become overloaded with resources and inaccessible to other users.

3. **DNS Reflection attacks**: cause responses that are significantly larger than requests to be sent directly to a victim by sending DNS requests to the target's source IP address. Attackers transmit forged request packets to a server with a changed source address in reflection-based flooding attacks [35]. Massive response packets are sent to the victim identified by a modified server source address because the server cannot tell the difference between legitimate and faked packets. An example of a reflection-based attack called an amplification-based flooding attack aims to trick the server into sending a large number of answer packets to a victim with few queries.

4. **HTTP Flood**: this happens when a web server is overloaded by HTTP requests, which sends an enormous volume of requests and is unable to handle valid requests.

5. **ICMP Flood**: occurs when an attacker depletes a victim's resources by bombarding the server with responses to a huge number of ICMP pings (echo requests and replies).

## Methodology

This section provides a comprehensive explanation of the dataset, attack detection architecture, algorithms, and data pre-processing steps used as well as details of the methodology administered.

## Impact of the behavior of traffic post attack

According to the type of topology, 100 packets are used to compute the entropy value, and a threshold value is chosen. A random value is computed based on the destination IP addresses. A flexible and quick way to calculate the standard distribution used as an anomaly detection tool was offered in the ahn entropy technique introduced in [36]. Here, we present entropy as a statistic for detecting DDoS attack traffic. Eq 1 calculates entropy.

$$Entropy(E) = \sum_{a=1}^{m} - R_a \log_2 R_a \tag{1}$$

In Eq 1, the approach entropy ($E$) is a function with probability $R_a$. To distinguish between classes that have to be trained, we need to know which characteristic in a set of training feature vectors is most helpful. The details are gained to reveal the significance of a certain feature vector characteristic. Entropy gained is depicted in Eq 2.

$$Gain(E, F1) = Entropy(E) - \sum_{c} \in features \frac{|E_c|}{E}$$

$$Entropy(E_c) \tag{2}$$

The number of values sampled is denoted as $E_c$ having a value $\in c$. A standard technique has been utilized to normalize the value **Gain** for every feature. Let *MGain* be the value for the Gain post performing the standard normalization. *MGain* is evaluated in Eq 3.

$$MGain(E, F1) = \frac{Gain(E, F1) - highest(Gain(E, F1))}{highest(Gain(E, F1) - lowest(Gain(E, F1)))} \tag{3}$$

Entropy sample space of flow length is identical to the post-attack entropy sample space.

We begin with the assumption that the entropy's flow length is $e(length)$ before the attack, then post-attack the flow length is $e_l(length)$ as shown in Eq 4.

$$e_l(length = 1) = -\sum_{a=b}^{\Phi} e(length = a) s_a^b r^b \log_2 R_a \tag{4}$$

In Eq 4, $e_l$ shows the value of entropy after the attack, $e$ denotes the value of entropy before the attack, the constant values are $s_a^b r^b$ and the length of the flows post-attack is $e_l(length = 1)$.

Let us assume that there are two lengths of flow $b_1$ and $b_2$, in case the same values are assigned for both the flow lengths, then the same values will be there for two flow lengths ($e(length = b_1) == e(length = b_2)$). The sample spaces $s_a^b r^b \log_2 R_a$ are defined as constant. Therefore, when the attack happens, before then the value of the two entropy's are not different. However, the values of entropy for the attack that happened after are not different, in other words ($e(length = b_1) == e(length = b_2)$).

To validate Lemma, the Mininet emulator (http://mininet.org/) is deployed for the Open Virtual Switch that was utilized for the switches in the network. Mininet operates on the Linux Operating System and we run our validation on UBUNTU. We note here that Mininet is primarily used for stationary network nodes. However, if you want to simulate the mobility patterns of vehicles in Mininet, you can achieve this by using the Mininet-WiFi extension. Mininet-WiFi extends Mininet to support wireless networks and mobility models, making it suitable for simulating vehicular networks.

The procedure for detection of attacks deploying entropy is based on normal traffic, where the randomness in 100 packets is very high, therefore, entropy is high. In the scenario of attacked traffic, since the maximum number of packets are directed to a unified host, randomness reduces which signifies entropy becomes very low. In this paper, the value of entropy is high, therefore, in case, the value exceeds the threshold, the detection of a DDoS attack plays an alarm in the SDN Ryu controller (https://ryu-sdn.org/).

## Degree of attack for DDoS attack identification

After a DDoS attack happens, DDoS identification in the SDN network begins. The detailed steps are described as follows: **Step 1**: The degree of the DDoS attack in SDN has four features $u_1$, $u_2$, $u_3$, $u_4$. $M(u_1)$, $M(u_2)$, $M(u_3)$, $M(u_4)$ denote the features of *MGain* values. Later, the degree of the attack (*SD*) is defined in the SDN network given in Eq 5.

$$SD = \frac{1}{m}\sum_{a=1}^{m}M(u_a); \forall m = 8 \tag{5}$$

**Step 2**: Let us assume that the flow is denoted as $U$, where $U_b$ signifies the *MGain* flow during time $b$, given in Eq 3. To identify the SDN attack, Eqs 3 and 5 are used. If $U_b = 0$ and $SD <= 1$, it means an attack is not identified, whereas if $U_b = 1$ and $SD >= 1$, it means an attack is identified. In case $SD > 1$, the flow passed is an attack during time $b$. Therefore, the network of SDN suffers from the attack, otherwise, the flow is not an attack at the same time. Algorithm 1 gives the details of the proposed approach.

**Algorithm 1** Algorithm for proposed approach
```
  Input: Degree of Attack
  Output: Recognization of DDoS
1: procedure BEGIN(:)
2:    if b ∈ B then
3:       Evaluate SD using Eq 5
4:    if SD ≥ 1
5:       DDoS attack identified on SDN network;
6:    else
7:       DDoS attack is not identified on SDN network;
8:    endif
9: endif
```

To differentiate DDoS attack packet flows from normal packet flows effectively, several sophisticated techniques and strategies can be employed. Firstly, monitoring traffic volume and rates is crucial; DDoS attacks typically generate a significantly higher volume of packets per second compared to normal traffic patterns. Analyzing these metrics allows for the detection of sudden spikes or anomalies that may indicate an ongoing attack. Secondly, statistical analysis plays a pivotal role in examining packet inter-arrival times and entropy [37]. DDoS attacks often exhibit irregular patterns and lower entropy due to the repetitive nature of attack packets, which contrasts with the more diverse and predictable patterns of normal traffic [38]. Thirdly, leveraging machine learning models, such as Random Forest, trained on historical data helps classify traffic based on learned patterns of normal versus attack behavior. These models can discern subtle deviations from established norms, enhancing the ability to detect and respond to emerging threats in real-time. Additionally, protocol and payload inspection techniques scrutinize network protocols and packet contents for anomalies indicative of attack signatures or malicious intent, providing deeper insights into the nature of traffic anomalies. By integrating these methods within a comprehensive detection framework, organizations can

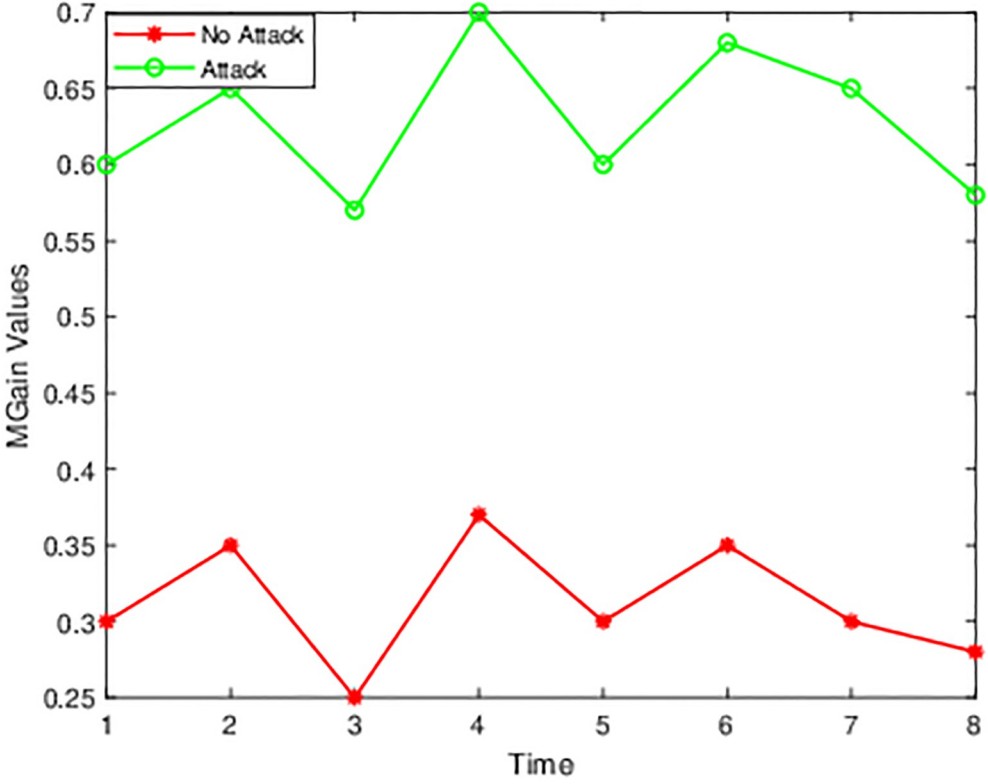

**Fig 3. Flow length.**

effectively differentiate DDoS attack packet flows from normal traffic, thereby fortifying their defenses and ensuring the resilience of their networks against evolving cyber threats.

Fig 3 shows the time and values for normal traffic and attacked traffic, indicating the *MGain* values for unique features in 8-time intervals. For the different time intervals, there is a fluctuation for different features. Fig 4 displays the changes that happen at various time points and features. Fig 4 shows that there are not many changes, and the MGain value for the flow arrival rate is higher than the flow length after the DDoS attack.

## Proposed approach

The **standard phase** is stated to be the state of the overall network. When the flow of messages increases, the network is assumed to be under attack, and the system injects into the **detection phase**. When the system reaches the detection phase, it must determine whether a DDoS is attacking the network or not. Whenever the escalated volume of communication extends a specific threshold, the detection phase is initiated. The **attack recognition phase** begins if a DDoS attack is detected in the system. The system looks to detect the attack's source and attack path during this stage. The system enters the **mitigation phase** after locating the attack source. All traffic coming from the attacker's source is halted during this period. Fig 5 depicts the system's change through its many stages. Each arrow indicates an event that enables the system to move between stages. A mitigation plan is put into action in the mitigation phase to stop attack flow. The suggested IoT DDoS defense mechanism is made up of several elements that each perform a different portion of the overall task.

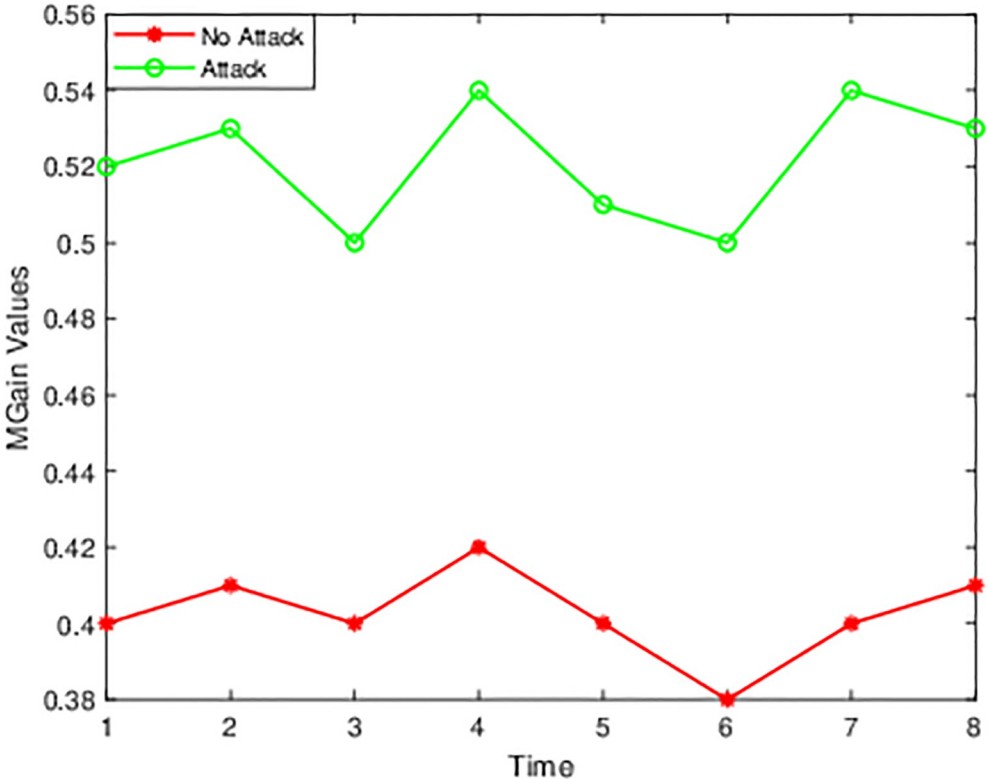

**Fig 4. Flow duration.**

The main elements of the framework are the frequency and traffic volume that allow the detection of any deviation from the network's typical behavior. Then, control is transferred to the detection stage if it detects a dramatic rise in the occurrence and size of messages attempting to reach the IoT gateway. A monitoring element in the detection stage is responsible for finding DDoS attacks in the IoT network. The monitoring element finds anomalies and verifies a DDoS attack when control is transferred to the detection stage. After that, the system enters the recognition stage. By analyzing data from the detection stage and utilizing the SDN's global view, the recognition stage can trace the attack path and identify the attacker. The system returns to an earlier stage if the attacker is not recognized. After recognizing the attacker, the Mitigation stage is initiated. A suitable defense tactic is employed in the mitigation stage to halt malicious traffic. After overcoming the attack, the system returns to the standard stage. All details are presented in Fig 5.

**Detection of attacks.** The detection module is the crucial subsystem in every DDoS security plan since it controls how proactive the system is. Due to the increased speed at which DDoS attacks are growing, the detection of DDoS attacks has become the main goal of this research work. Most DDoS detection methods include statistics, data mining, machine learning, computational intelligence, and/or knowledge-based approaches. Many newer proposed systems combine many of these elements. A monitoring component is part of a DDoS detection mechanism that scans the network for any deviation from usual activity before determining whether or not a DDoS attack is the culprit. The monitoring component notifies the system or network administrator if it finds a potential DDoS attack. Two different approaches make up the Detection Phase of the SDN-IoT framework: (1) analysis of the system and detect

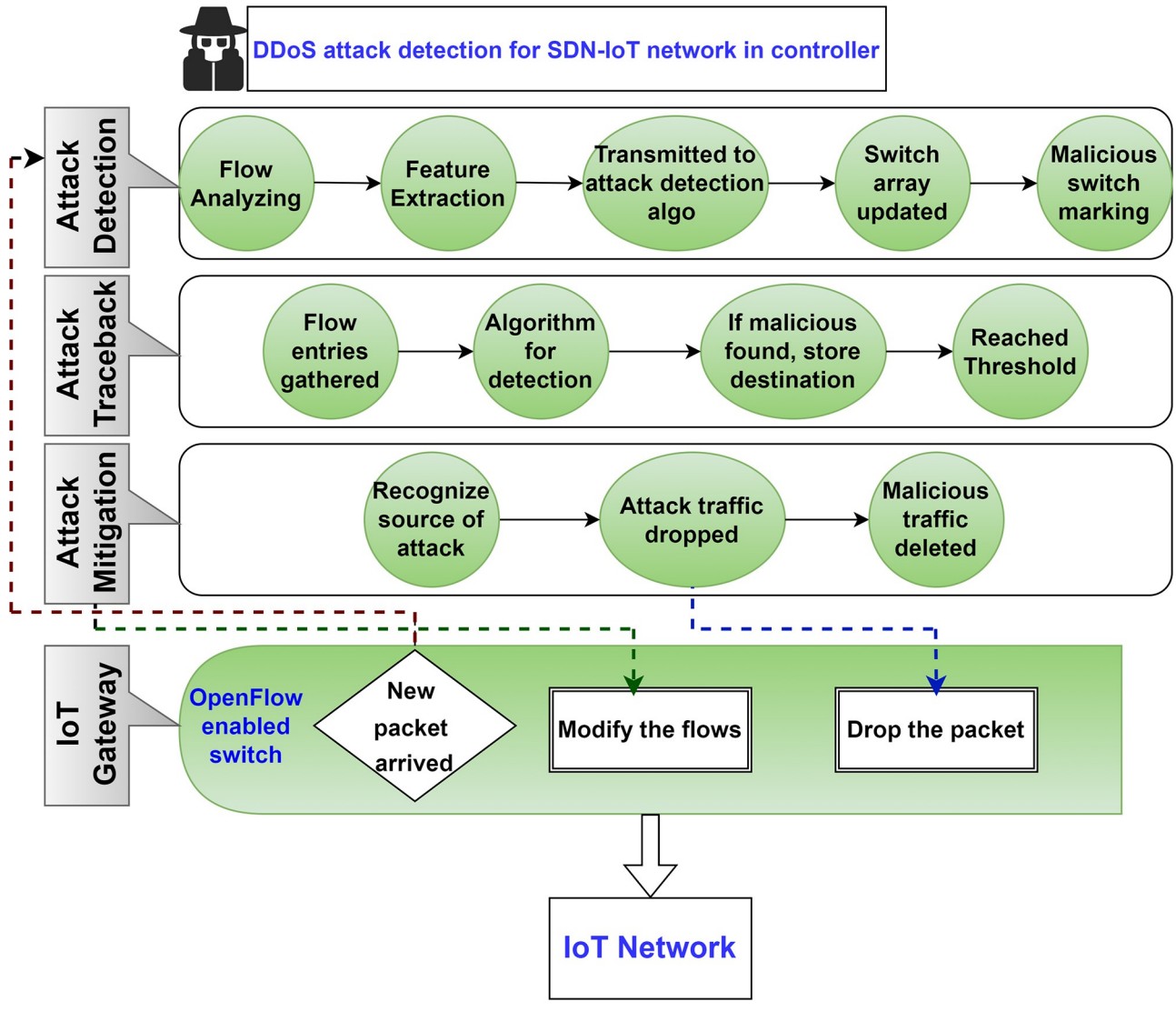

**Fig 5. Framework of proposed SDN-IoT for attack detection.**

the unexpected flow of messages, and (2) evaluate abnormal behavior and verify the DDoS attack. The rate at which messages arrive at the IoT gateway is computed in the first sub-module, and the massive data abnormal detection method is utilized to find the outliers of messages.

Algorithm 2 shows the precise steps involved in DDoS attack detection. The amount of new messages designated as **Flag** ($f$) is increased by one each time a new **message** ($m$) arrives. The modular split of Flag and **Threshold** ($t$) is computed, a specified upper bound on the number of new messages. If the remainder is not zero, the controller receives the new message and processes it. Instead, the time is specified. The gap between the **present time** ($e_curr$) **and the previous time** ($e_prev$), when the residual for the modular division of flag and threshold was zero, is used to calculate the amount of time that has **elapsed** ($e$). The newest **message's rate** ($r$) is determined by dividing the threshold by the passing time.

**Algorithm 2** Attack Detection

```
Input: Arrived newest message = m
Output: Detection of malicious behavior
1: procedure BEGIN(:)
2:    Increment the flag by 1:= f++
3:  if f mod t = 0 then:
4:    e = e_curr - e_prev
5:    r = t/e
6:  else
7:    transmits the newest messages to the SDN controller
8:  endif
9:  if r is benign, then
10:    inform the SDN controller
11:  else
12:    Confirm whether the malicious has happened due to DDoS
13:  endif
```

To determine whether the outlier discovered is part of a DDoS attack, irregularity must be carefully analyzed in the detection stage. Algorithm 2 is used to describe the latter portion of the detection stage. ML algorithms are employed in the proposed framework to find the potential DDoS attack. ML algorithms can discover patterns and identify trends from inadequate or complex data by extracting the necessary elements. The data from the flow entries is retrieved from the controller and then sent to the trained neural network or ML algorithms after the monitored sub-module has identified the abnormality. ML algorithms identify malicious and DDoS-based traffic. Before deploying a neural network model for authentic detection, it must first be trained. Using features of the malicious traffic, a dataset that was built in advance is used for training. A distinct set of numbers is needed to demonstrate malicious and benign traffic in a dataset.

It is crucial to emphasize that the system is independent to facilitate the efficiency of the detection and mitigation operations. Therefore, no human intervention is necessary even though the system generates an alarm to alert the network administrator when a DDoS attack is detected. Fig 6 describes a schematic diagram that illustrates how the proposed system operates. The SDN controller exports IP flow measurements or features every second via the OpenFlow protocol. These measurements consist of heterogeneous data that can be divided into qualitative and quantitative features, such as source/destination ports and IP addresses. Quantitative features include package rate and bits per second. The qualitative measurement needs to be changed into a quantitative one for the detection stage to be able to use this data.

In Fig 7, as soon as the system is started, the neural network begins to be trained. ML algorithms take the characteristics of both malicious and benign traffic as input. These figures are compared with the anomaly discovered, aiding in the identification of a DDoS attack. The packet-count that each flow entry matches, the flow entry time and the rate of each flow entry are the attributes that ML algorithms use as input. The features listed can change based on the level of accuracy desired and are derived from the controller's flow data [39]. These attributes are used to generate the eigenvalues for the ML algorithms, which aid in distinguishing between benign and malicious communications. The destination address is found and recorded in a list named malicious IP list after a malicious flow entry is found. A DDoS alarm is triggered and the controller halts the execution of flow statistics messages if malicious flow entries rise to a threshold number. If the flow entry is benign and the largest number of malicious entries has not been reached, the following flow entry is evaluated.

The proposed approach ensures accurate differentiation between reconnaissance, DoS, and DDoS traffic by addressing the issues of unbalanced datasets and overfitting in the deployment of ML algorithms specifically designed for intrusion detection in SDN environments enhances the model's ability to learn and recognize complex attack patterns [40]. These algorithms are

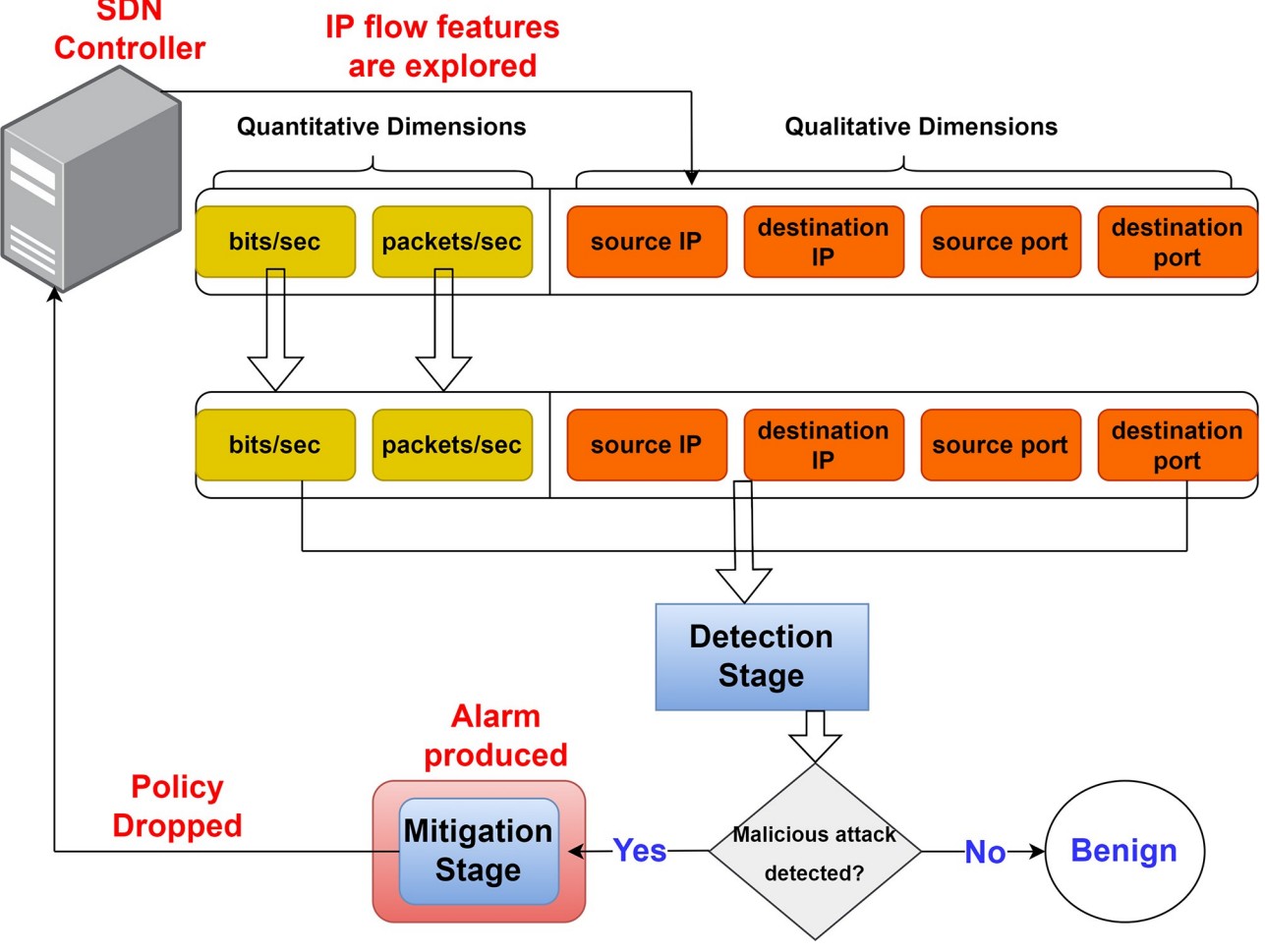

**Fig 6. SDN security system.**

trained to differentiate between reconnaissance, DoS, and DDoS attacks by learning from labeled examples. The approach explicitly tackles the issue of unbalanced datasets, which is common in cybersecurity datasets where benign traffic vastly outnumbers malicious traffic. Techniques such as oversampling the minority class, undersampling the majority class, or using synthetic data generation (e.g., SMOTE) can be employed to balance the dataset, ensuring that the ML model is exposed to sufficient examples of all types of traffic.

## Limitations of selection of features for model evaluation

There exist various potential restrictions when evaluating a model for identifying DDoS assaults in SDVN by using a limited portion of the gathered characteristics. These limitations may affect the results' generalizability. (i.) By selecting only a subset of features, important information that could be crucial for distinguishing between different types of traffic (e.g., benign, reconnaissance, DoS, DDoS) might be omitted. This can lead to a decrease in the model's ability to accurately identify all types of malicious activities. (ii.) When a limited number of features are used, the model might not capture the complexity of the data, leading to underfitting. Underfitting occurs when the model is too simple to capture the underlying patterns in

**Fig 7. Traffic categorization.**

the data, resulting in poor performance across different datasets. (iii.) A model trained on a limited set of features may lack robustness to variations and anomalies in real-world traffic. This can make the model less effective in adapting to new types of attacks or changes in traffic patterns that were not represented in the training data.

## Results and discussions

### Experimental setup

Mininet is an emulator for utilizing huge networks on the restricted resources of a unified computer or any virtual machine (http://mininet.org/). Mininet has been used to enable the research in SDN and hence, it permits the users to run unmodified code on the virtual hardware or PC.

Jupyter Notebook [41] was used throughout the entire experiment on an HP laptop running the Windows 10 64-bit operating system (https://jupyter.org/). An Intel i7 $8^{th}$ generation Quad-core processor with a processing unit and 16.0 GB of RAM was used. To load and pre-

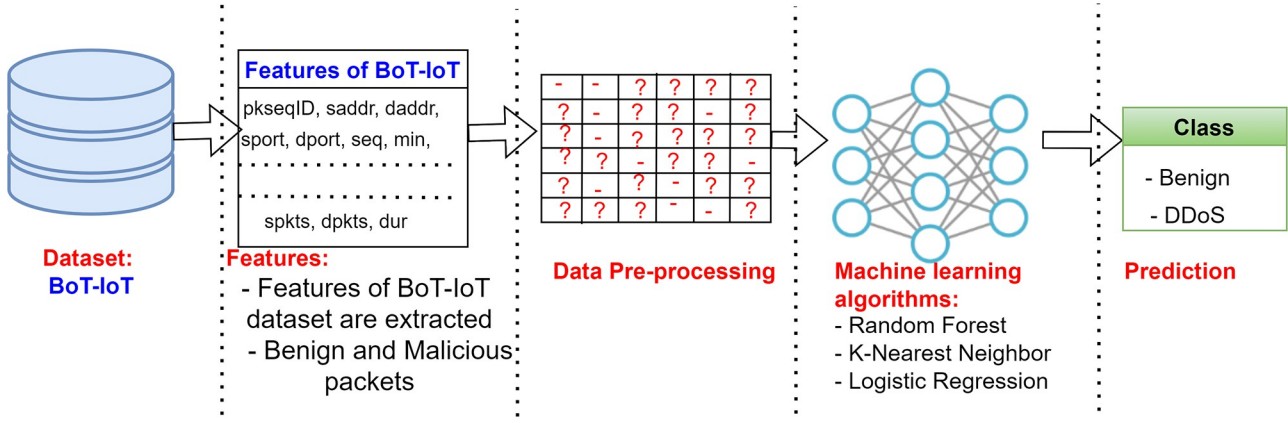

**Fig 8. Proposed Model for categorization of attacks in SDN-IoT using ML.**

process the dataset, Pandas and the NumPy library were utilized. Scikit-learn has been used for evaluating the model, evaluation metrics to check the effectiveness and training-testing data analysis. Data visualization was carried out using the Matplotlib module. The layout of the proposed model for the categorization of attacks in SDN-IoT using ML is shown in Fig 8.

## Dataset

The effectiveness of the threat detection methodology is substantially impacted by choosing an appropriate dataset. As per our study, BoT-IoT [42] is utilized for threat detection in IoT. Several datasets have been used by various authors for intrusion detection in IoT contexts. The BoT-IoT dataset created by The University of New South Wales (UNSW), Canberra, Australia, was the source of the data employed in this study. The complete dataset has 72 million records and includes both benign traffic and several attack traffic types like DDoS, DoS, Data exfiltration, OS, and Service Scan. 74 different files make up the BoT-CSV IoT's [34] for the SDN version. For such experimental analysis, the traffic for two attacks—DDoS and reconnaissance—as well as benign traffic was taken into consideration for efficiency of management.

 **Characterstics of BoT-IoT dataset**.

- **Origin and Purpose**: The BoT-IoT dataset is derived from the publicly available IoT traffic dataset called "Bot-IoT 2018" and is designed for evaluating intrusion detection systems (IDS) in IoT environments. It contains traffic data captured from various IoT devices and networks, including simulated attack scenarios.

- **Dataset Size and Scope**: The exact size of the BoT-IoT dataset can vary depending on the specific subset used for experimentation. Typically, it consists of a substantial amount of network traffic data, including both benign and malicious traffic samples.

- **Types of Attacks**: The dataset covers a diverse range of attacks commonly encountered in IoT environments. These include various forms of DoS (Denial of Service), DDoS, reconnaissance attacks, and other malicious activities targeting IoT devices and networks.

- **Traffic Diversity**: It includes traffic generated by different IoT devices and applications, each characterized by unique communication patterns and protocols. This diversity reflects real-world IoT environments, making it suitable for testing the robustness of intrusion detection systems under various conditions.

- **Anomaly and Attack Scenarios**: The BoT-IoT dataset incorporates both normal traffic patterns and anomalous behaviors that simulate attacks. This allows researchers and practitioners to assess the effectiveness of detection algorithms in accurately distinguishing between benign activities and potential security threats.

- **Dataset Usage**: Researchers often use subsets or specific versions of the BoT-IoT dataset for experimental evaluations in intrusion detection, machine learning-based classification, anomaly detection, and other related studies aimed at enhancing IoT security.

## Feature extraction

The synchronization of transmitting and acquiring packets, as well as the subsequent automatic development of labeling these packets as benign or malicious, make it difficult to capture network data while guaranteeing the labeling process. We created some scripts on the Cron Linux features over the UBUNTU Tap VM to complete this work. An explicit benign or malicious scenario is required to be performed when the programs run at a specific time. For instance, when the DDoS was being generated, we planned the deployment of customized bash scripts that used `hping3` and `golden-eye` to perform the DDoS malicious attacks. Simultaneously, background traffic was generated normally. `Tshark` [43, 44] is the tool that was running concurrently to collect raw packets and save them in 1 GB PCAP files to make it easier to retrieve network information. Therefore, defining the IP addresses of the attacker and recipient workstations allowed us to distinguish between benign and malicious traffic while trying to ensure that only attacking traffic would be sent between both groups.

There is a total of 46 features in the BoT-IoT dataset (the bottom three features are labeled), out of which 22 features are extracted for our research as shown in Table 1.

Furthermore, the categories `saddr`, `sport`, `daddr`, `dport`, and `proto` are regarded as network flow identifiers because they can each be used to uniquely identify a flow at a given time and aid in labeling. For such dataset training and validation of machine learning models using binary classification, malicious occurrences are labeled with a 1 and benign occurrences with a 0.

## Data pre-processing

The data pre-processing stage is the essential step in the evaluation. Certain proposed framework attributes, including `saddr`, `daddr`, and `proto` (see Table 1), are of the categorized

**Table 1. Extracted features from BoT-IoT dataset.**

| Sno. | Feature Name | Description | Sno. | Feature Name | Description |
|---|---|---|---|---|---|
| 1 | pkseqID | Identifier of tuple | 12 | mean | Average record duration for all records |
| 2 | proto | Descriptive illustration of the network flow's transaction protocols | 13 | $N_I N_{Conn} P_{DstIP}$ | Number of connections coming into each destination IP |
| 3 | saddr | IP address of the source | 14 | drate | Packets per second from destination to source |
| 4 | sport | Port number of source | 15 | srate | Packets per second from source to destination |
| 5 | daddr | IP address of the destination | 16 | max | Maximum record duration for data collected |
| 6 | dport | Port number of destination | 17 | dur | A complete-duration record |
| 7 | seq | The sequence number of tool | 18 | spkts | Count of packets from source to destination |
| 8 | stddev | Combined records standard deviation | 19 | dpkts | Count of packets from destination to source |
| 9 | $N_I N_{Conn} P_{SrcIP}$ | Number of connections from each source IP | 20 | attack | Labeling the class: Benign-0 and malicious-1 |
| 10 | min | Minimum record duration for data collected | 21 | category | Category related to traffic |
| 11 | state_number | Depiction of feature state numerically | 22 | subcategory | Sub-category related to traffic |

type and must be transformed into algorithm executable form. Source and destination IP addresses are indicated by `saddr` and `daddr`, respectively, while the protocol type used during the flow is indicated by `proto`. This feature is converted into the integer type. Every source and destination IP address has been given a number. There are a total of 301 IP addresses used for the BoT-IoT data set.

A well-known issue in ML is imbalanced data, which happens when the dispersion of the various classes is biased. The distribution of the various classes in an unbalanced data set can be somewhat unbalanced or extremely unbalanced. Any learning model trained using a wildly unbalanced dataset would perform poorly in terms of predicting outcomes for minor classes. The BoT-IoT dataset has approx. 0.15% of benign data. Additionally, the BoT-IoT data set contains roughly 55% of DDoS and roughly 50% of DoS data. As a result, this dataset cannot be utilized to train and forecast benign, DDoS, or DoS packets alone. There are 34 million DoS packets and 40 million DDoS packets in the BoT-IoT data collection. In the proposed framework that has been presented, we divide the 3.2 benign packets into 14 equal data chunks for DDoS and DoS packets. There are 3.2 million distinct DDoS and DoS packets for all of the 14 chunks, meaning that there are 3.2 million benign packets, 3.2 million DDoS packets, and 3.2 million DoS packets in each chunk. Consequently, all of the 14 chunks include 7.5 million packets combined. With this approach, the over-fitting issue is reduced and the data for the three classes are distributed equally among all chunks.

## Defining machine learning model classes

A system can learn and get better based on experience in ML, a subfield of Artificial Intelligence (AI), without any need for external programming [45]. Algorithms or techniques are used in ML to build learning models from data. The main goal of ML is to make it possible for computers to learn automatically, without human intervention, so they can make future decisions that are accurate. The choice of $K$-Nearest Neighbor, Random Forest, and Logistic Regression classifiers for this study in the context of SDN Intrusion Detection for IoT environments is based on their distinct characteristics and strengths that make them suitable for different aspects of intrusion detection. Here are the reasons for selecting these classifiers, along with their respective advantages and disadvantages:

1. **KNN**: Data is categorized using KNN, a fundamental and simple machine learning technique, according to similar distance metrics. In this situation, the distance may be of the Manhattan or Euclidean types. The value of $k$ is chosen once the proximity between the data points is computed. The data points with the closest distance are given the same class, as well as the value of $k$ can be any integer value. With a rise in the frequency of nearest neighbors, the model's accuracy rises (i.e., the value of $k = 5$). Since the KNN algorithm makes no assumptions about the data, it is non-parametric, which is useful when dealing with legitimate data.
   **Advantages**:

   - Performs well on small datasets where the computational cost is manageable.

   - Can handle multi-class classification without any modification.

   - KNN is a lazy learner, meaning it requires no explicit training phase, making it fast to deploy.
   **Disadvantages**:

   - High computational cost during prediction as it involves calculating the distance between the query point and all points in the dataset.

- Requires storing the entire dataset, which can be impractical for large datasets.

- Performance can be degraded by irrelevant or redundant features, necessitating careful feature selection and normalization.

2. **RF**: Random Forest is built on the idea of ensemble learning, which combines different classifiers to handle complicated data problems and enhance the model's overall effectiveness. RF is made up of several distinct decision trees that analyze various dataset fragments. The voting from every tree is considered, and the majority votes are used to forecast the outcome. The problem of overfitting can be avoided by increasing accuracy performance by increasing the number of trees in RF.
   **Advantages**:

- Generally provides high accuracy and robustness to overfitting due to averaging multiple trees.

- Can handle both classification and regression tasks and works well with high-dimensional data.

- The ensemble nature reduces the risk of overfitting compared to individual decision trees.
  **Disadvantages**:

- More complex and computationally intensive than single decision trees, particularly during the training phase.

- While individual trees are interpretable, the overall Random Forest model is less transparent.

- Can be slower to train, especially with large datasets and a high number of trees.

3. **LR**: It is a technique for estimating different values using a set of independent variables as input. The algorithm's anticipated result is a dependent categorical variable. It aids in the prediction of a probabilistic value between 0 and 1. Linear Regression and LR are comparable, but LR differs in the kind of problem that needs to be solved. When compared to other ML algorithms, LR is important since it can categorize a variety of data types and identify the most relevant variable.
   **Advantages**:

- Computationally efficient and fast to train, making it suitable for real-time applications.

- Provides coefficients that are easily interpretable, indicating the strength and direction of the relationship between features and the outcome.

- Effective for binary classification tasks, which are common in intrusion detection.
  **Disadvantages**:

- Assumes a linear relationship between the features and the log odds of the outcome, which may not capture complex, non-linear patterns in IoT traffic.

- May perform poorly if the underlying data relationships are non-linear and complex.

- Can be sensitive to outliers, which can skew the model's performance.

## Validation metrics

Performance analysis is a crucial step after data preparation and ML model training. Various performance indicators based on the confusion matrix are used to assess the effectiveness of

the machine learning model. Before applying the model to previously unexplored data, it is crucial to assess its overall effectiveness.

**Confusion metric.** A performance evaluation method for machine learning classification in SDN-IoT is the confusion matrix. With the aid of the testing data's real values, an evaluation is performed. Confusion Matrix assists in ML performance metric calculation and error detection (FP and FN).

- **True Positives (TP)**: are a true positive label that the classifier determines to be positive.

- **True Negatives (TN)**: are negative labels that the classifier recognizes as true.

- **False Positives (FP)**: are produced when a valid negative label is mistakenly interpreted as a positive.

- **False Negatives (FN)**: are when a real positive label is mistakenly interpreted as negative by the classifier.

A positive event is typically viewed in the context of cyber-security research to be a malicious occurrence, and the accurate classification of an event is considered to be a true positive outcome. A negative event is a good thing, and the accurate description is true negativity. An inaccurate classification could lead to the labeling of a benign event as malicious. It is thought that this misclassification is a false positive. Similarly, it is considered a false negative when a hostile event is labeled as a benign occurrence.

The metrics used for the performance evaluation in our study are:

1. **Accuracy**: This statistic aids in figuring out a classifier's accuracy. It establishes how many accurate predictions the model built. It is the proportion between the number of accurate predictions and all of the model's other predictions. Eq 6 is evaluated as:

$$Accuracy = \frac{TP + TN}{TP + TN + FP + FN} \tag{6}$$

2. **Precision**: It is the percentage of correctly predicted values to all correctly predicted values as determined by the classification model. Eq 7 is evaluated as:

$$Precision = \frac{TP}{TP + FP} \tag{7}$$

3. **Recall**: It describes the overall number of records for a given class that can be correctly predicted using whatever data is available. Eq 8 is evaluated as:

$$Recall = \frac{TP}{TP + FN} \tag{8}$$

4. **F1-Score**: also referred to as the mean of Recall and Precision, employs Recall and Accuracy to evaluate the model thoroughly. Eq 9 is computed as:

$$F1 - Score = \frac{2 * TP}{2 * TP + FP + FN} \tag{9}$$

**Table 2. Categories with its number of samples.**

| Categories | Samples |
|---|---|
| DDoS | 1,541,315 |
| DoS | 1,320,148 |
| Reconnaissance | 72,919 |
| Benign | 370 |
| Theft | 65 |

**Table 3. Categories with its number of samples.**

| Label | Samples |
|---|---|
| Malicious | 2,934,447 |
| Benign | 370 |

5. **False Positive and False Negative**: **False Positives** are instances where legitimate traffic is incorrectly classified as a DDoS attack. **False Negatives** are instances where actual DDoS attack traffic is incorrectly classified as legitimate.

## Performance analysis

The model is trained in the first experiment of this study employing the test/train split functions from the Scikit Learn library on the BoT-IoT dataset. Because the dataset only has around

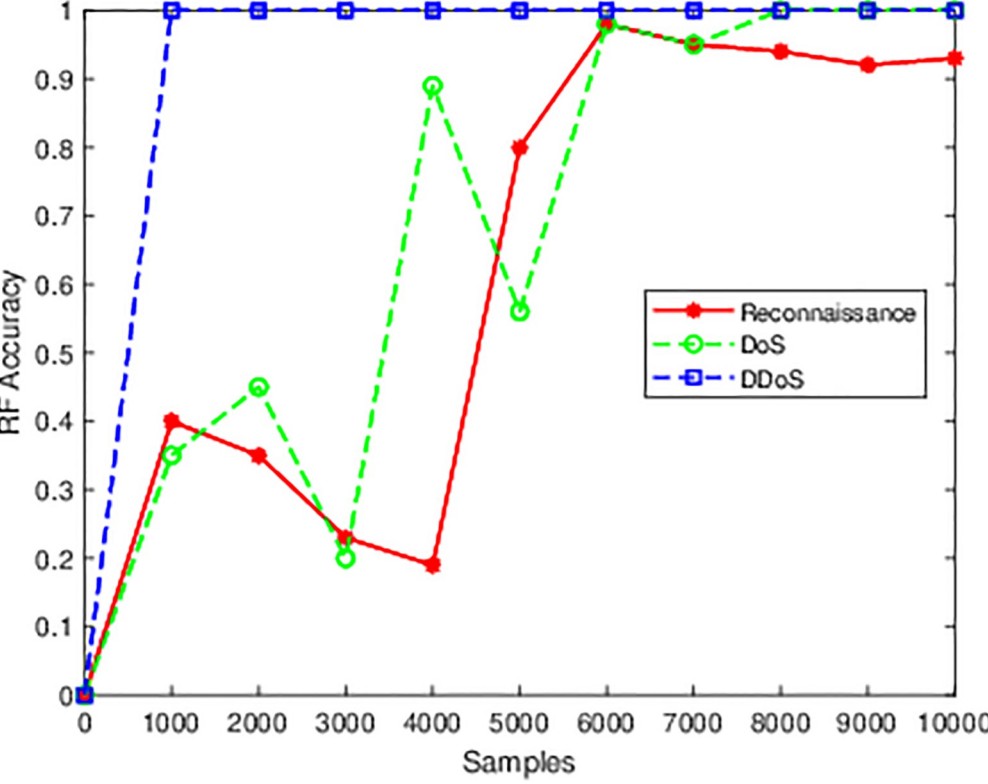

**Fig 9. RF model.**

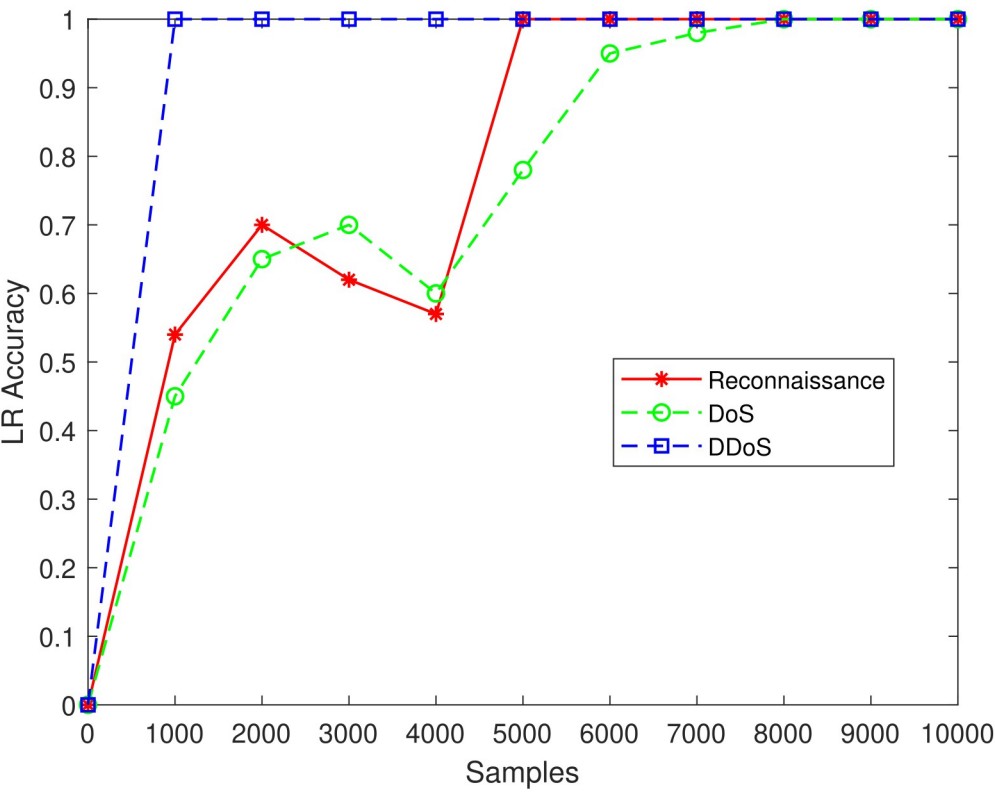

**Fig 10. LR model.**

72 million records, two attacks shown in Tables 2 and 3: a DDoS attack with 1, 541, 315 million samples, a DoS attack with 1,320,148, a reconnaissance attack with about 72, 919, Benign data with 370 and theft with 65, respectively were chosen for this research to make handling it easier. KNN, RF, and LR are the ML classifiers utilized in this research.

Statistical flow measurements involve collecting and analyzing various metrics that describe the behavior of network traffic. Common metrics include:

**Packet Count**: Number of packets transmitted over a period. **Byte Count**: Amount of data (in bytes) transmitted over a period. **Flow Duration**: The time duration of a flow from the first to the last packet. **Packet Inter-arrival Time**: Time intervals between consecutive packets. **Flow Rate**: The rate at which packets/bytes are transmitted.

These metrics help in understanding the patterns and anomalies in network traffic. For instance, a sudden spike in packet count or flow rate can indicate a potential attack.

**Criteria for Determining the Best Sample Size** Determining the best sample size for the dataset and evaluating the balance between sample size and model performance involves several critical steps and criteria including: 1. Evaluate key performance metrics such as accuracy, precision, recall, and F1-score. These metrics provide a comprehensive view of the model's ability to correctly identify different types of traffic, including benign and malicious traffic (reconnaissance, DoS, DDoS). 2. Consider the training time and computational resources required. Larger sample sizes typically improve model performance but also increase the time and resources needed for training. 3. Ensure that the sample size is large enough to achieve statistically significant results. Small sample sizes may lead to high variance in performance metrics and unreliable conclusions.

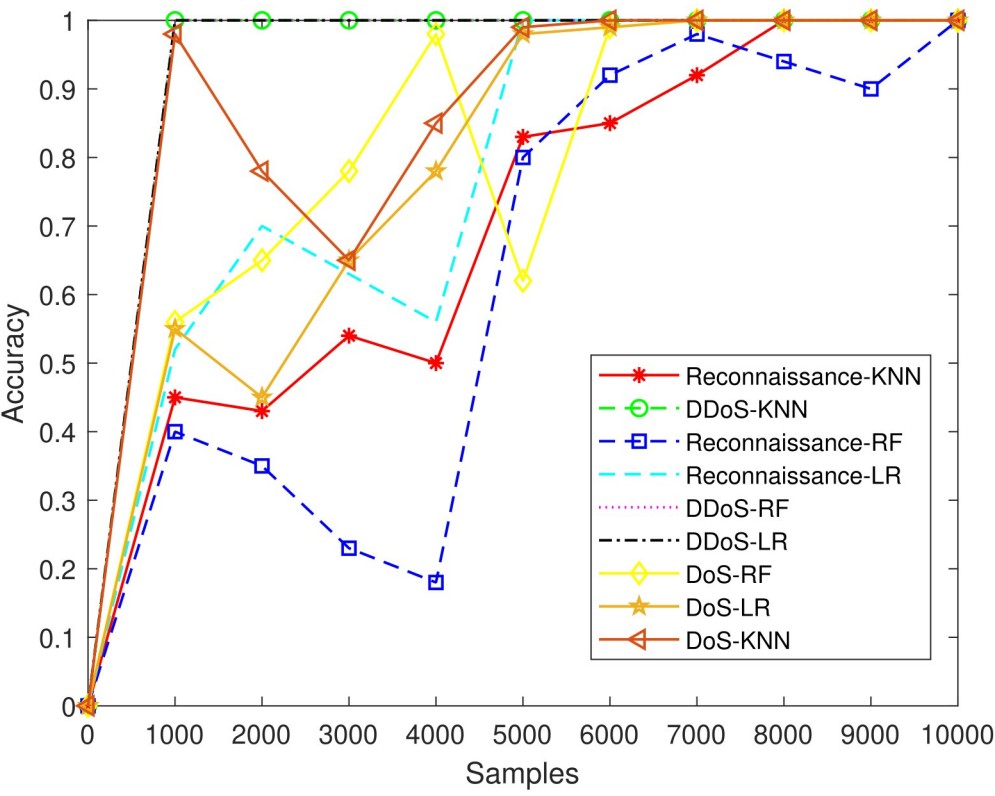

**Fig 11. Comparative analysis of all attacks.**

By using a large number of records with a very unbalanced class distribution, the model was able to perform well. This experiment includes a small subset of samples from the full dataset to determine the threshold sample limit at which the model performs at its best. The BoT-IoT dataset contains an unusually high proportion of malicious samples compared to benign samples, which can strain resources like memory and processing time. Identical to the last experiment, this one took into account three attacks: DDoS, DoS, and reconnaissance. The BoT-74 IoT's CSV files are initially concatenated into a single file. The concatenated file is used to extract the aforementioned malicious traffic as well as benign traffic. Following this is feature selection pre-processing, which selects a limited subset of data for deployment: 10, 000 and 1, 000 samples, respectively, of malicious and benign traffic. The subset of samples is next subjected to pre-processing processes. By manually inputting the number of training samples for every iteration, a sequence of ten iterations is carried out. The iteration samples were selected using a procedure of trial and error. The Threshold/Breakpoint value is considered to be the first occurrence of the greatest value during 5 repetitions. There are two sections to this research. The number of benign samples is determined in the first section, and then malicious samples are ordered in a series for 5 iterations. For DDoS, DoS and reconnaissance threats, the model has been trained using three classifiers: KNN, FR, and LR. Here is a description of the threshold comparison analysis for the research's initial phase: Fig 9 displays the KNN classifier's threshold evaluation for reconnaissance, DoS and DDoS attacks. 500 fixed benign samples are taken into account for the KNN following the trial-and-error procedure. At 6000 samples, the model achieves an 85% Accuracy in reconnaissance, indicating a pre-breakthrough point for the threshold value. For reconnaissance, a threshold of 99% is reached after 8000 malicious

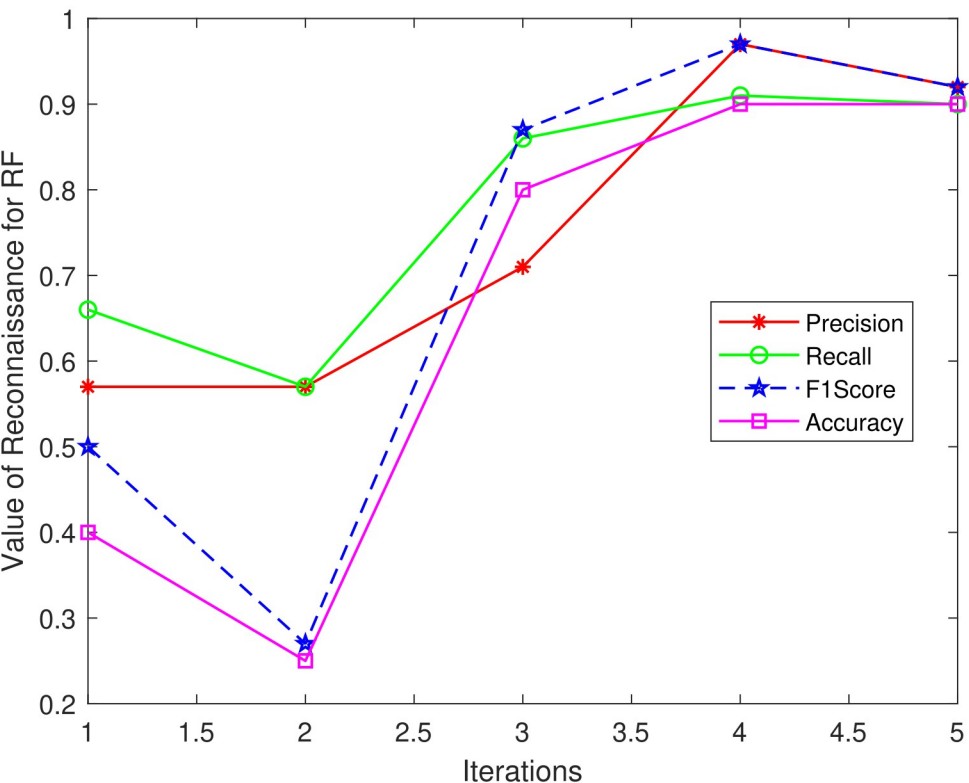

**Fig 12. RF_Reconnaissance attack.**

samples, and then for DDoS, a threshold of 100% is reached at 1000 samples. In contrast to DDoS, where a perfect score is attained with 1000 attack samples, the classifier accuracy in this figure shows a progressive upward trend forward toward the threshold value for reconnaissance. The DoS samples have reached 78% with 5000 samples. A comparison of the three attacks reveals that DDoS needs fewer attack samples to reach the threshold value.

Fig 9 shows the Random Forest classifier's threshold analysis. The maximum classifier score that was attained using the same quantity of fixed benign samples (500 samples) as KNN was roughly 50%. As a result, new fixed benign samples had to be generated using the second round of trial and error. For RF, 58 benign samples were used as a fixed value, and similarly to KNN, 5 iterations of series changes were made to the attack samples. Employing 6000 assault samples for reconnaissance, the iterations that use the improved benign sample were able to hit the 97% threshold. At 1000 samples, DDoS reached a threshold of 100%. For DoS with 6000 samples, the threshold reaches 98% similar to the reconnaissance. Just after threshold iteration, a very slight percentage decline is seen for reconnaissance, however, this was not the case for the DDoS attack and DoS attack. As with KNN, DDoS and DoS require only a few attack samples to produce an accurate classifier.

Fig 10 shows the threshold analysis for Logistic Regression, in which the same 500 fixed benign samples were collected as for KNN. According to the graph, reconnaissance can reach a threshold of 99% with 5000 attack samples whereas DDoS can do so with just one. In contrast to DDoS attacks, a tiny drop in classifier scores is seen in reconnaissance after the threshold values. By contrasting the two forms of attacks in LR, it can be seen that DDoS needs fewer samples to reach the threshold.

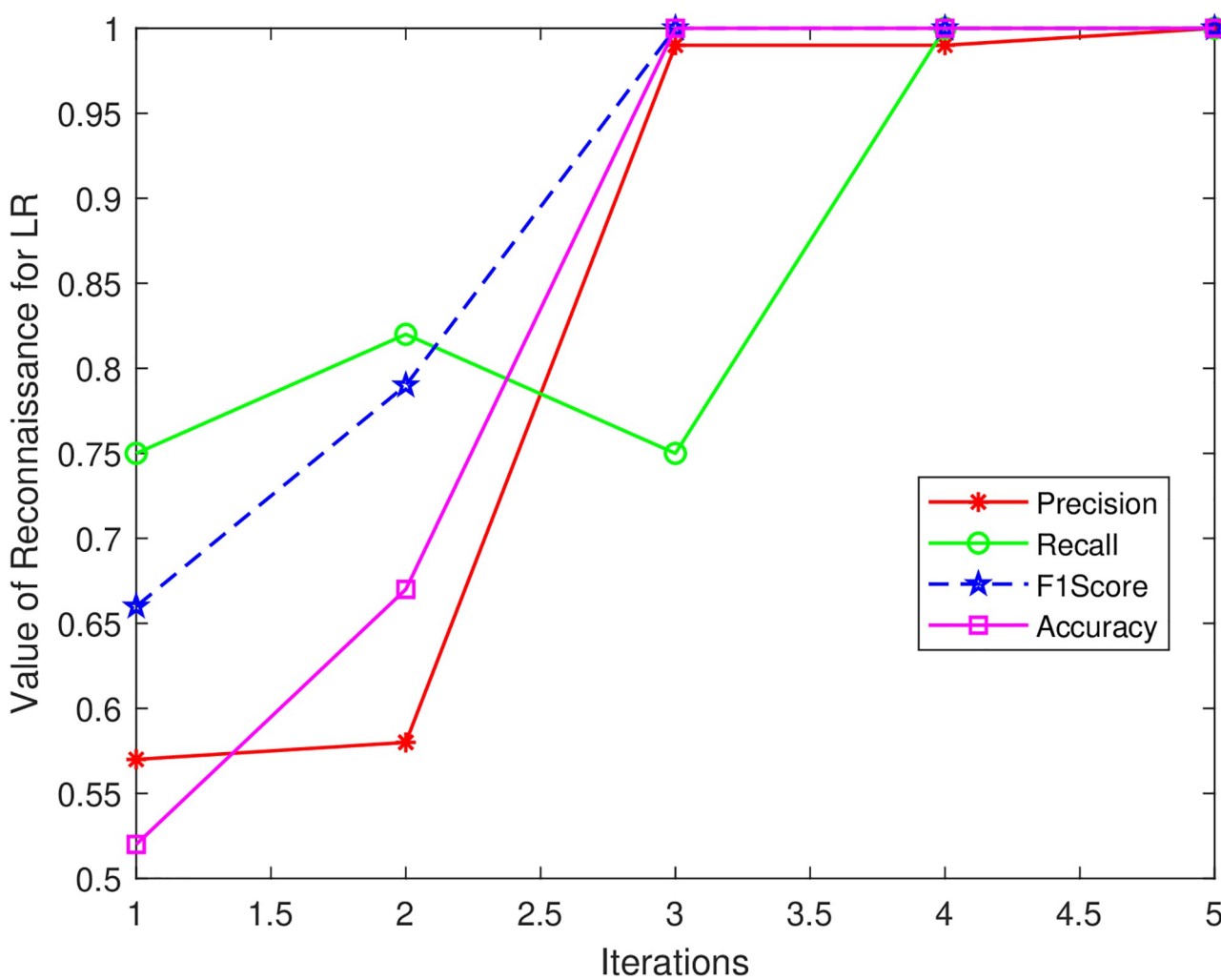

**Fig 13. LR_Reconnaissance attack.**

Fig 11 is a composite threshold analysis for two attacks employing all three classifiers. To reach the criteria for reconnaissance, LR needed 5000 samples, then for RF and KNN, respectively, 6000 and 8000 samples, respectively. In the initial iteration, LR successfully crossed the threshold of 100% for DDoS and DoS attacks. KNN and RF displayed comparable behaviours and met the criteria at 1000 sample attacks. This experiment has shown that DDoS takes fewer resources overall. To meet the criterion for KNN, RF, and LR, more samples must be collected than during reconnaissance.

The same is illustrated visually in Figs 12–14 where it can be seen that efficiency for all classification models begins to improve after iteration 5. KNN and LR both obtain 100% for all metrics in iterations 4 and 5, however, RF achieves the maximum Precision and F1-score of 92%, Accuracy of 91%, and Recall of 90% at Iteration 5. These findings demonstrate a match between the performance measure values and the threshold value calculated from the classifier score.

In Fig 15, the feature named sport has dropped 6% than the value of the threshold. The rest of the features show the least change from the threshold value. The features sport and dport have dropped by 24% than the value allocated as the threshold. The other features

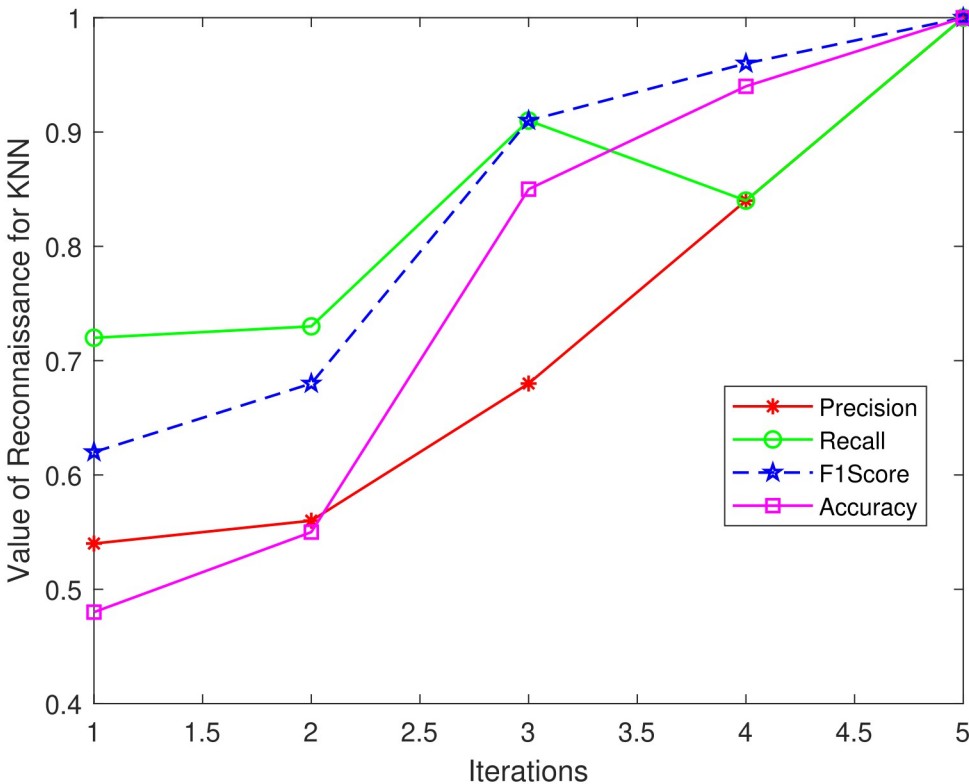

**Fig 14. KNN_Reconnaissance attack.**

indicate the count of packets is equivalent to the threshold which shows the attack rate is less in the features extracted.

Fig 16 shows the RF model performance on real-time imbalance data and the class balance dataset is good. The Accuracy and ROC AUC curve can be done on the BoT-IoT imbalanced data i.e. 99.6% and 99.2%, respectively and from balanced data 92.1% and 92.2%, respectively. For the BoT-IoT dataset, the Accuracy result from RF is good. This indicates that the RF classifier is an effective algorithm in the botnet detection system.

The effectiveness of the proposed approach in detecting different types of DDoS attacks beyond those included in the BoT-IoT dataset hinges on its adaptability and generalization capabilities. The use of ML algorithms, such as KNN, Random Forest, and Logistic Regression, enables the system to learn patterns and characteristics of DDoS attacks from the training data and apply this knowledge to identify new and varied attack patterns. The system can be designed to continuously update its models with new data, including previously unseen attack types. This allows the ML algorithms to adapt to evolving threat landscapes. Beyond predefined attack types, the system can employ anomaly detection techniques to flag any traffic patterns that deviate significantly from the learned normal behavior, allowing for the detection of novel or sophisticated attacks.

## Conclusion

This study proposed an SDN-based IoT in vehicular networks for the identification of DDoS attacks using attributes from the most recent benchmark dataset, BoT-IoT. Researchers from the University of New South Wales generated the dataset. By choosing an equal number of

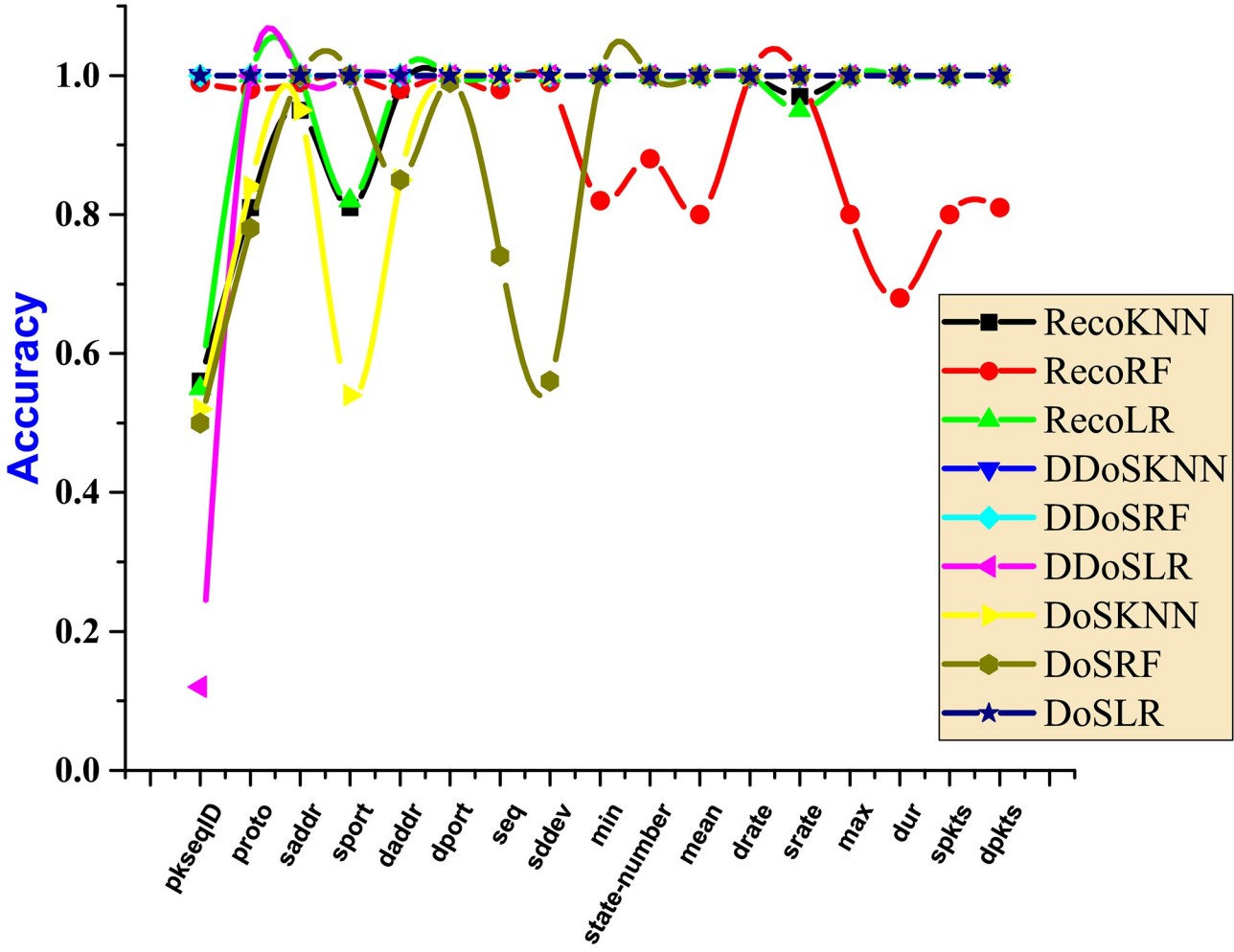

**Fig 15. Dropping extracted features for all types of attacks.**

packets from each category, the issues with the public data set, such as mismatch in nature and over-fitting, are addressed. By using the ML technique, we were able to classify benign, reconnaissance, DoS, and DDoS traffic with an Accuracy of 91% and represent the extracted features in the dataset. This work is distinctive in that it covers the most recent benchmarked data set while reducing (nearly by half) the number of features provided for the recognition of malicious traffic. By utilizing additional well-known and benchmark data sets in the future, we want to further enhance the feature comparison and selection technique. There are various potential paths for further investigation. Investigating the effectiveness of the ML model using various subset combinations is one area in which this research could be extended.

## Future scope

The future scope could explore enhancing detection accuracy and response speed in real-time scenarios as vehicular networks become increasingly complex and data-intensive. Advanced machine learning models, including deep learning techniques, could be further developed to detect even subtle and adaptive DDoS attack patterns. Additionally, integrating blockchain technology may offer secure and decentralized data sharing, strengthening the reliability of detection methods. Expanding the research to address attacks on autonomous vehicles, and

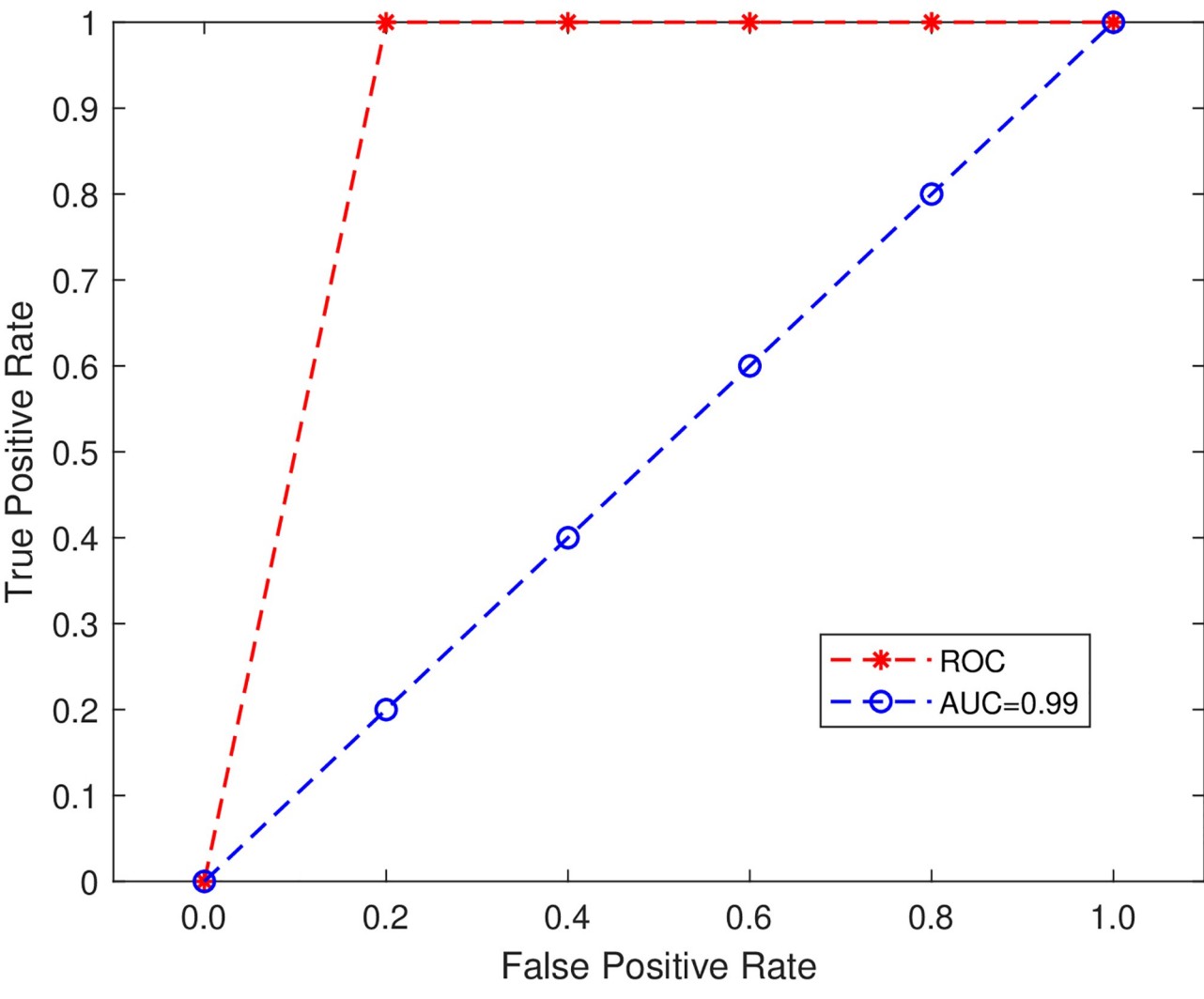

**Fig 16. ROC curve.**

exploring edge computing for distributed attack mitigation, could also pave the way for more resilient and responsive vehicular network security systems.

## Author Contributions

**Data curation:** Shalli Rani.

**Formal analysis:** Shalli Rani.

**Funding acquisition:** Maha Driss.

**Methodology:** Himanshi Babbar.

**Project administration:** Maha Driss.

**Visualization:** Himanshi Babbar.

**Writing – original draft:** Himanshi Babbar.

**Writing – review & editing:** Shalli Rani.

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
