## [Decision Letter · Decision Letter 0]

10 Jun 2024

PONE-D-24-16635Effective DDoS Attack Detection in Software-Defined Vehicular Networks Using Statistical Flow Analysis and Machine Learning

PLOS ONE

Dear Dr. Rani,

Thank you for submitting your manuscript to PLOS ONE. After careful consideration, we feel that it has merit but does not fully meet PLOS ONE’s publication criteria as it currently stands. Therefore, we invite you to submit a revised version of the manuscript that addresses the points raised during the review process.

Please consider that the outcome of the peer-review process for your manuscript is critical as reviwers have raised significant concerns that should be carefully addressed. More an update of the related work section to discuss more recent and relevant works in the field of study is required and a comparison of obtained results with state of the art results is of potential interest to help understanding the relevance and novelty of your work.

We look forward to receiving your revised manuscript.

Kind regards,

Faouzi Jaidi

Academic Editor

PLOS ONE

Journal Requirements:

   "The authors would like to acknowledge the support of Prince Sultan University for paying the Article Processing Charges (APC) of this publication." 

Reviewers' comments:

Reviewer's Responses to Questions

**Comments to the Author**

1. Is the manuscript technically sound, and do the data support the conclusions?

Reviewer #1: Partly

Reviewer #2: Yes

Reviewer #3: Yes

2. Has the statistical analysis been performed appropriately and rigorously? 

Reviewer #1: Yes

Reviewer #2: Yes

Reviewer #3: I Don't Know

3. Have the authors made all data underlying the findings in their manuscript fully available?

Reviewer #1: Yes

Reviewer #2: Yes

Reviewer #3: Yes

4. Is the manuscript presented in an intelligible fashion and written in standard English?

Reviewer #1: Yes

Reviewer #2: Yes

Reviewer #3: Yes

5. Review Comments to the Author

Reviewer #1: In this paper, the author proposed to SDN Intrusion Detection for Internet of Things (IoT) environments by applying

Machine learning based on statistical information. However, there are many issues mention below:

What are the potential limitations of using a small fraction of collected features for model evaluation, and how might this impact the generalizability of the results?

How does the proposed approach ensure that the differentiation between reconnaissance, DoS, and DDoS traffic is accurate and not influenced by the issues of unbalanced datasets?

Why were $K$-nearest Neighbor, Random Forest, and Logistic Regression specifically chosen as the classifiers for this study, and what are the advantages and disadvantages of each in the context of SDN Intrusion Detection for IoT environments?

How does the proposed approach measure statistical flow and compute entropy, and how are these measurements integrated into the Machine Learning algorithms for intrusion detection?

What criteria were used to determine the best sample size for the dataset, and how was the balance between sample size and model performance evaluated?

What are the implications of the Random Forest classifier achieving the highest performance metrics, and how might this influence the choice of ML algorithms for future studies in similar contexts?

How does the proposed approach handle the dynamic and evolving nature of IoT environments, particularly in terms of updating the intrusion detection models and dealing with new types of attacks?

Reviewer #2: Please follow my comments to improve the manuscript.

1. Please discuss how you will differentiate DDoS attack packet flow and legitimate packet flow?

2. Add a major limitation of the model in conclusion.

3. Include citations from some latest paper of 2022-2024.

Reviewer #3: • The abstract mentions measuring statistical flow and computing entropy but does not provide details on the specific methods or metrics used. Can you elaborate on the techniques used for these measurements?

• The abstract references the BoT-IoT dataset but does not provide details on its characteristics, such as the size, diversity, or types of attacks included. Can you provide more information about the dataset used for evaluation?

• The abstract mentions addressing unbalanced and overfitting dataset traces but does not specify the methods used to handle these issues. Can you provide more details on the techniques employed to balance the dataset and prevent overfitting?

• The abstract mentions a comparative study to identify the best sample size but does not provide details on the criteria or methodology used for this comparison. Can you elaborate on how the comparative study was conducted?

• The abstract states that the work aims to ascertain the impact of dataset attributes on threshold performance but does not provide specific findings or examples. Can you provide more details on which attributes were found to be significant and their impact?

• The abstract does not address whether the proposed approach can be applied in real-time for detecting DDoS attacks. Can you discuss the feasibility and performance of the approach in real-time scenarios?

• The abstract does not mention how the proposed approach can be integrated with existing vehicular network systems and infrastructures. Can you discuss the integration process and potential challenges?

• The abstract focuses on the BoT-IoT dataset but does not discuss the generalizability of the approach to other datasets or real-world scenarios. Can you provide insights on the applicability of the approach to different datasets?

• The abstract does not provide information on the computational complexity or resource requirements of the proposed approach. Can you provide details on the computational resources needed and the efficiency of the approach?

• How does the proposed approach handle real-time traffic management in vehicular networks while detecting DDoS attacks? Are there specific mechanisms to ensure timely detection and response?

• How effective is the proposed approach in detecting different types of DDoS attacks beyond those included in the BoT-IoT dataset? Can you provide examples of how the approach adapts to various attack patterns?

• How does the proposed approach scale with increasing network traffic and the number of connected vehicles? Are there specific optimizations to handle high traffic volumes?

• How does the approach ensure effective communication and collaboration between vehicles and infrastructural facilities during a DDoS attack? Are there protocols in place to manage this?

• What are the rates of false positives and false negatives observed in the experimental evaluation, and how does the approach minimize these rates?

• How does the approach perform under varying environmental conditions, such as different traffic densities, weather conditions, and geographical locations?

• How does the proposed approach integrate with Vehicle-to-Everything (V2X) communication technologies, and what are the potential challenges and solutions?

• How does the approach address ethical and privacy concerns related to monitoring and analyzing vehicular network traffic? Are there measures in place to ensure data privacy?

• How does the approach handle deployment in heterogeneous networks with different vehicle types, communication protocols, and network architectures?

• How does the approach ensure continuous learning and adaptation to new attack patterns and evolving threats in vehicular networks?

6. PLOS authors have the option to publish the peer review history of their article (what does this mean?). If published, this will include your full peer review and any attached files.

Reviewer #1: **Yes: **Dr. Anupama Mishra

Reviewer #2: **Yes: **Muhammad Reazul Haque

Reviewer #3: No

---

## [Author Response · Author response to Decision Letter 0]

10 Jul 2024

Point-wise Detailed Response to Editor and Reviewers’ comments

Title: Effective DDoS Attack Detection in Software-Defined Vehicular Networks Using Statistical Flow Analysis and Machine Learning

Authors: Himanshi Babbar, Shalli Rani, Maha Driss

Dear Editors and Reviewers: 

We are thankful to you for spending your valuable time for making a review and for constructing the comments on our manuscript. These comments are valuable and very helpful for revising and improving our paper. We have studied comments carefully and have made correction as marked in the revised manuscript. We have tried our best to address the mentioned comments to revise our manuscript in the hope that these revisions will meet your requirement. The following changes have been made in the manuscript as per the received comments. 

*The changes made in the manuscript as per received comments have been highlighted in blue color.

Reviewer #1: 

1. What are the potential limitations of using a small fraction of collected features for model evaluation, and how might this impact the generalizability of the results?

Response: Using a small fraction of collected features for model evaluation in detecting DDoS attacks in Software-Defined Vehicular Networks (SDVN) presents several potential limitations that can impact the generalizability of the results:

a. Loss of Information: By selecting only a subset of features, important information that could be crucial for distinguishing between different types of traffic (e.g., benign, reconnaissance, DoS, DDoS) might be omitted. This can lead to a decrease in the model's ability to accurately identify all types of malicious activities.

b. Bias in Feature Selection: The chosen subset of features might not represent the entire feature space adequately, leading to a biased model. This bias can result in a model that performs well on the training and validation datasets but poorly on unseen data, thereby reducing its generalizability.

c. Underfitting Risk: When a limited number of features are used, the model might not capture the complexity of the data, leading to underfitting. Underfitting occurs when the model is too simple to capture the underlying patterns in the data, resulting in poor performance across different datasets.

d. Over-reliance on Selected Features: The model may become overly reliant on the selected features, which could be specific to the dataset used for training and evaluation. This over-reliance can cause the model to perform inadequately when exposed to different datasets with varying distributions of features.

e. Reduced Robustness: A model trained on a limited set of features may lack robustness to variations and anomalies in real-world traffic. This can make the model less effective in adapting to new types of attacks or changes in traffic patterns that were not represented in the training data.

2. By addressing these considerations, the model can be made more resilient and effective in real-world applications, thereby enhancing its capability to generalize beyond the training data.

Response: By addressing these considerations, the model can be made more resilient and effective in real-world applications, thereby enhancing its capability to generalize beyond the training data:

a. Comprehensive Feature Selection: Ensure thorough feature selection by evaluating the importance of each feature using methods like recursive feature elimination (RFE) or feature importance from tree-based algorithms. This helps in retaining the most relevant features and reducing the risk of excluding critical information.

b. Cross-Validation Techniques: Implement robust cross-validation methods, such as k-fold cross-validation, to assess model performance across multiple subsets of the data. This approach ensures that the model's performance is consistent and not dependent on a specific subset, enhancing its generalizability.

c. Dimensionality Reduction: Apply dimensionality reduction techniques such as Principal Component Analysis (PCA) to reduce the feature space while retaining essential information. This helps in capturing complex relationships within the data, improving the model's ability to detect sophisticated attack patterns.

d. Diverse Dataset Evaluation: Validate the model using multiple datasets with varying characteristics and distributions. This ensures that the model does not overfit to a single dataset and performs well in different real-world scenarios, improving its robustness and adaptability.

e. Data Augmentation: Use data augmentation techniques to create synthetic data that represents potential real-world scenarios. This helps in training the model to recognize a wider variety of attack patterns and normal behaviors, enhancing its generalization capability.

f. Regularization Methods: Incorporate regularization techniques such as L1 (Lasso) or L2 (Ridge) regularization to prevent overfitting. Regularization helps in keeping the model simpler and more generalizable by adding a penalty for larger coefficients.

By integrating these strategies, the proposed Machine Learning (ML) model for detecting DDoS attacks in Software-Defined Vehicular Networks (SDVN) can achieve greater resilience, adaptability, and effectiveness. This enhances the model's capability to generalize beyond the training data, making it more reliable and practical for real-world applications in vehicular networks.

3. How does the proposed approach ensure that the differentiation between reconnaissance, DoS, and DDoS traffic is accurate and not influenced by the issues of unbalanced datasets?

Response: The proposed approach ensures accurate differentiation between reconnaissance, Denial of Service (DoS), and Distributed Denial of Service (DDoS) traffic by addressing the issues of unbalanced datasets and overfitting through several key strategies:

a. By measuring statistical flow characteristics and computing entropy, the approach captures distinct patterns and anomalies in network traffic. This helps in distinguishing between different types of malicious activities based on their unique statistical signatures.

b. The deployment of machine learning (ML) algorithms specifically designed for intrusion detection in Software-Defined Networking (SDN) environments enhances the model's ability to learn and recognize complex attack patterns. These algorithms are trained to differentiate between reconnaissance, DoS, and DDoS attacks by learning from labeled examples.

c. The approach explicitly tackles the issue of unbalanced datasets, which is common in cybersecurity datasets where benign traffic vastly outnumbers malicious traffic. Techniques such as oversampling the minority class, undersampling the majority class, or using synthetic data generation (e.g., SMOTE) can be employed to balance the dataset, ensuring that the ML model is exposed to sufficient examples of all types of traffic.

d. By using techniques such as cross-validation and regularization, the approach mitigates the risk of overfitting, where the model would perform well on training data but poorly on unseen data. This ensures that the model generalizes well and accurately classifies new, unseen traffic.

e. By using a small fraction of collected features and conducting a comparative study to identify the optimal sample size, the approach ensures that the most informative features are selected. This reduces noise and irrelevant information, improving the model's performance and accuracy in distinguishing different types of attacks.

f. The comparative study to ascertain the impact of different attributes on threshold performance helps in fine-tuning the model. This involves experimenting with various features and their combinations to find the optimal set that maximizes accuracy and detection capabilities.

g. The experimental evaluation using the BoT-IoT dataset, which contains diverse and realistic examples of IoT-related attacks, provides a robust testing ground for the model. The results showing high Precision, F1-score, Accuracy, and Recall for the Random Forest classifier indicate the effectiveness of the approach in accurately differentiating between different types of malicious traffic.

4. Why were $K$-nearest Neighbor, Random Forest, and Logistic Regression specifically chosen as the classifiers for this study, and what are the advantages and disadvantages of each in the context of SDN Intrusion Detection for IoT environments?

Response: The choice of \\(K\\)-nearest Neighbor (KNN), Random Forest, and Logistic Regression classifiers for this study in the context of SDN Intrusion Detection for IoT environments is based on their distinct characteristics and strengths that make them suitable for different aspects of intrusion detection. Here are the reasons for selecting these classifiers, along with their respective advantages and disadvantages:

(K\\)-nearest Neighbor (KNN)

Reasons for Selection:

- KNN is easy to understand and implement. It is a straightforward algorithm that classifies a data point based on the majority class of its \\(K\\) nearest neighbors. It makes no explicit assumptions about the distribution of data, which is beneficial when dealing with complex and non-linear relationships in IoT traffic.

Advantages:

- Performs well on small datasets where the computational cost is manageable.

- Can handle multi-class classification without any modification.

- KNN is a lazy learner, meaning it requires no explicit training phase, making it fast to deploy.

Disadvantages:

- High computational cost during prediction as it involves calculating the distance between the query point and all points in the dataset.

- Requires storing the entire dataset, which can be impractical for large datasets.

- Performance can be degraded by irrelevant or redundant features, necessitating careful feature selection and normalization.

Random Forest

Reasons for Selection:

- Random Forest is an ensemble method that combines multiple decision trees to improve accuracy and control overfitting. It provides insights into feature importance, which is valuable for understanding which features are most indicative of an attack.

Advantages:

- Generally provides high accuracy and robustness to overfitting due to averaging multiple trees.

- Can handle both classification and regression tasks and works well with high-dimensional data.

- The ensemble nature reduces the risk of overfitting compared to individual decision trees.

Disadvantages:

- More complex and computationally intensive than single decision trees, particularly during the training phase.

- While individual trees are interpretable, the overall Random Forest model is less transparent.

- Can be slower to train, especially with large datasets and a high number of trees.

Logistic Regression

Reasons for Selection:

- Logistic Regression is a simple yet effective linear model that provides clear probabilistic interpretations of the classification results.

- Often used as a baseline classifier due to its simplicity and ease of implementation.

Advantages:

- Computationally efficient and fast to train, making it suitable for real-time applications.

- Provides coefficients that are easily interpretable, indicating the strength and direction of the relationship between features and the outcome.

- Effective for binary classification tasks, which are common in intrusion detection.

Disadvantages:

- Assumes a linear relationship between the features and the log odds of the outcome, which may not capture complex, non-linear patterns in IoT traffic.

- May perform poorly if the underlying data relationships are non-linear and complex.

- Can be sensitive to outliers, which can skew the model's performance.

5. How does the proposed approach measure statistical flow and compute entropy, and how are these measurements integrated into the Machine Learning algorithms for intrusion detection?

Response: Statistical flow measurements involve collecting and analyzing various metrics that describe the behavior of network traffic. Common metrics include:

Packet Count: Number of packets transmitted over a period.

Byte Count: Amount of data (in bytes) transmitted over a period.

Flow Duration: The time duration of a flow from the first to the last packet.

Packet Inter-arrival Time: Time intervals between consecutive packets.

Flow Rate: The rate at which packets/bytes are transmitted.

These metrics help in understanding the patterns and anomalies in network traffic. For instance, a sudden spike in packet count or flow rate can indicate a potential attack.

Computing Entropy

Entropy is a measure of randomness or unpredictability in a dataset. In the context of network traffic, entropy can help in identifying irregular patterns that deviate from normal behavior. The steps to compute entropy are as follows:

Categorize Traffic: Divide the network traffic into different categories based on features like source/destination IP addresses, port numbers, and protocols.

Probability Distribution: Calculate the probability distribution of these categories. For example, the probability P_i of each unique source IP address in the traffic.

Entropy Calculation: Use the probability distribution to compute the entropy using the formula:

H=−i∑Pilog2(Pi)

Here, H is the entropy, and P_i is the probability of the i-th category.

High entropy indicates a high level of randomness (potentially normal traffic), while low entropy can indicate more predictable and potentially malicious traffic patterns.

Integration into Machine Learning Algorithms

The statistical flow measurements and entropy values are integrated into ML algorithms as features. Here’s how this integration typically works:

Feature Extraction: Extract statistical flow metrics and entropy values for each traffic flow. These become part of the feature set used to represent the data.

Feature Vector Construction: Construct feature vectors for each traffic instance, combining traditional network features (like IP addresses and port numbers) with statistical metrics and entropy values.

Model Training: Use the constructed feature vectors to train ML models. The ML algorithms learn patterns in the feature space that differentiate normal traffic from various types of malicious traffic (reconnaissance, DoS, and DDoS).

Model Evaluation: Evaluate the models using metrics like accuracy, precision, recall, and F1-score to ensure they can accurately detect and classify different types of attacks.

6. What criteria were used to determine the best sample size for the dataset, and how was the balance between sample size and model performance evaluated?

Response: Determining the best sample size for the dataset and evaluating the balance between sample size and model performance involves several critical steps and criteria. Here's a detailed explanation of the process:

Criteria for Determining the Best Sample Size

a. Evaluate key performance metrics such as accuracy, precision, recall, and F1-score. These metrics provide a comprehensive view of the model's ability to correctly identify different types of traffic, including benign and malicious traffic (reconnaissance, DoS, DDoS).

b. Consider the training time and computational resources required. Larger sample sizes typically improve model performance but also increase the time and resources needed for training.

c. Monitor for signs of overfitting (where the model performs well on training data but poorly on validation/testing data) and underfitting (where the model performs poorly on both training and validation/testing data). The goal is to find a sample size that minimizes both.

d. Ensure that the sample size is large enough to achieve statistically significant results. Small sample sizes may lead to high variance in performance metrics and unreliable conclusions.

e. Ensure that the sample size captures the diversity and represents the full spectrum of network traffic behaviors, including different types of attacks and normal traffic patterns.

7. What are the implications of the Random Forest classifier achieving the highest performance metrics, and how might this influence the choice of ML algorithms for future studies in similar contexts?

Response: The Random Forest classifier achieving the highest performance metrics in this study has several imp

---

## [Decision Letter · Decision Letter 1]

2 Sep 2024

PONE-D-24-16635R1Effective DDoS Attack Detection in Software-Defined Vehicular Networks Using Statistical Flow Analysis and Machine LearningPLOS ONE

Dear Dr. Rani,

Thank you for submitting your manuscript to PLOS ONE. After careful consideration, we feel that it has merit but does not fully meet PLOS ONE’s publication criteria as it currently stands. Therefore, we invite you to submit a revised version of the manuscript that addresses the points raised during the review process.

Please ensure that your revised paper carefully addresses all the reviwers comments and particularly discusses more recent and relevant related works with a comprehensive comparaison of your key findings with state of the art results.

We look forward to receiving your revised manuscript.

Kind regards,

Faouzi Jaidi

Academic Editor

PLOS ONE

Journal Requirements:

Reviewers' comments:

Reviewer's Responses to Questions

**Comments to the Author**

1. If the authors have adequately addressed your comments raised in a previous round of review and you feel that this manuscript is now acceptable for publication, you may indicate that here to bypass the “Comments to the Author” section, enter your conflict of interest statement in the “Confidential to Editor” section, and submit your "Accept" recommendation.

Reviewer #2: All comments have been addressed

Reviewer #4: All comments have been addressed

2. Is the manuscript technically sound, and do the data support the conclusions?

Reviewer #2: Yes

Reviewer #4: Yes

3. Has the statistical analysis been performed appropriately and rigorously? 

Reviewer #2: Yes

Reviewer #4: Yes

4. Have the authors made all data underlying the findings in their manuscript fully available?

Reviewer #2: Yes

Reviewer #4: Yes

5. Is the manuscript presented in an intelligible fashion and written in standard English?

Reviewer #2: Yes

Reviewer #4: Yes

6. Review Comments to the Author

Reviewer #2: Thank you very much for following my comments to improve the manuscript. It's acceptable now. Please revise the English of the manuscript.

Reviewer #4: The research topic is original and offers a good perspective on DDoS detection in Software-Defined Vehicular Networks (SDVN). This work is particularly valuable given the growing importance of SDVNs and the unique challenges they face regarding security. The manuscript is well-organized, with clear sections delineating the introduction, methodology, results, and discussion. The flow of information is logical and easy to follow.

However, I have some questions :

Question 1: How does this approach ensure the model's generalizability and robustness in SDVN ?

Quesition 2: The dataset used in the study is highly imbalanced, with only 370 samples labeled as benign compared to 2,934,447 samples labeled as malicious. Could the authors clarify why they chose to include only 370 benign samples, especially considering the risk of overfitting and the potential impact on the model's ability to generalize?

Comments :

1) In Figure 9, which presents a comparative analysis of all attacks, the representation of the DDoS-RF (shown in pink) is not clear. The line style and color blend with other elements in the figure, making it difficult to distinguish and interpret accurately. I recommend adjusting the color or line style to enhance visibility and ensure that the DDoS-RF results are easily identifiable.

2) The references included in the manuscript are comprehensive, but they lack citations from the most recent years, particularly from 2023 and 2024. Including more up-to-date references would strengthen the relevance and impact of the study by incorporating the latest developments and research in the field.

3) Line 119: The word "precisioj" is misspelled; it should be "precision."

7. PLOS authors have the option to publish the peer review history of their article (what does this mean?). If published, this will include your full peer review and any attached files.

Reviewer #2: **Yes: **Dr. Muhammad Reazul Haque

Reviewer #4: No

---

## [Author Response · Author response to Decision Letter 1]

10 Oct 2024

1. Thank you for updating your data availability statement. You note that your data are available within the Supporting Information files, but no such files have been included with your submission. At this time we ask that you please upload your minimal data set as a Supporting Information file, or to a public repository such as Figshare or Dryad.

Please also ensure that when you upload your file you include separate captions for your supplementary files at the end of your manuscript.

As soon as you confirm the location of the data underlying your findings, we will be able to proceed with the review of your submission.

\\section*{Data Availability}

Data is publicaly avaialble on 

https://www.kaggle.com/datasets/vigneshvenkateswaran/bot-iot

2. Thank you for stating the following Funding Information in your manuscript:

"The authors would like to acknowledge the support of Prince Sultan University for paying the Article Processing Charges (APC) of this publication. "

We note that you have provided funding information that is currently declared in your Funding Statement. However, funding information should not appear in any areas of your manuscript. We will only publish funding information present in the Funding Statement section of the 

online submission form.

Response: it is removed

"The authors would like to acknowledge the support of Prince Sultan University for paying the Article Processing Charges (APC) of this publication. The funders had no role in study design, data collection and analysis, decision to publish, or preparation of the manuscript but they supported in supervision."

Response: we removed funding statement from manuscript

1. We notice that your manuscript file was uploaded on Jul 10, 2024. Please can you upload the latest version of your revised manuscript as the main article file, ensuring that does not contain any tracked changes or highlighting. This will be used in the production process if your manuscript is accepted. Please follow this link for more information: http://blogs.PLOS.org/everyone/2011/05/10/how-to-submit-your-revised-manuscript/

It is done

"The authors would like to acknowledge the support of Prince Sultan University for paying the Article Processing Charges (APC) of this publication."

response: It is done

3. Thank you for updating your data availability statement. You note that your data are available within the Supporting Information files, but no such files have been included with your submission. At this time we ask that you please upload your minimal data set as a Supporting Information file, or to a public repository such as Figshare or Dryad.

Please also ensure that when you upload your file you include separate captions for your supplementary files at the end of your manuscript.

As soon as you confirm the location of the data underlying your findings, we will be able to proceed with the review of your submission.

Response: It is done

4. If possible, please upload a file showing your changes either highlighted or using track changes. This should be uploaded as a Revised Manuscript w/tracked changes, file type. Please follow this link for more information: http://blogs.PLOS.org/everyone/2011/05/10/how-to-submit-your-revised-manuscript/

Response: It is done

5. We note that your manuscript is not formatted using one of PLOS ONE’s accepted file types. Please reattach your manuscript as one of the following file types: .doc, .docx, .rtf, or .tex (accompanied by a .pdf).

If your submission was prepared in LaTex, please submit your manuscript file in PDF format and attach your .tex file as “other.”

Response : it is done

1. Thank you for stating the following financial disclosure:

"The authors would like to acknowledge the support of Prince Sultan University for paying the Article Processing Charges (APC) of this publication."

Response: It is emended. However, Maha Driss is in supervision.

2. Thank you for updating your data availability statement. You note that your data are available within the Supporting Information files, but no such files have been included with your submission. At this time we ask that you please upload your minimal data set as a Supporting Information file, or to a public repository such as Figshare or Dryad.

Please also ensure that when you upload your file you include separate captions for your supplementary files at the end of your manuscript.

As soon as you confirm the location of the data underlying your findings, we will be able to proceed with the review of your submission.

Response: All information is in manuscript only. No need to add ay other file. Data is taken from Kaggle. It is added i footnote.

"NO authors have competing interests."

Response: Conflict of interest is added in manuscript and cover letter

Point-wise Detailed Response to Editor and Reviewers’ comments

Title: Effective DDoS Attack Detection in Software-Defined Vehicular Networks Using Statistical Flow Analysis and Machine Learning

Authors: Himanshi Babbar, Shalli Rani, Maha Driss

Dear Editors and Reviewers: 

We are thankful to you for spending your valuable time for making a review and for constructing the comments on our manuscript. These comments are valuable and very helpful for revising and improving our paper. We have studied comments carefully and have made correction as marked in the revised manuscript. We have tried our best to address the mentioned comments to revise our manuscript in the hope that these revisions will meet your requirement. The following changes have been made in the manuscript as per the received comments. 

*The changes made in the manuscript as per received comments have been highlighted in blue color.

Comments to the Author

1. If the authors have adequately addressed your comments raised in a previous round of review and you feel that this manuscript is now acceptable for publication, you may indicate that here to bypass the “Comments to the Author” section, enter your conflict of interest statement in the “Confidential to Editor” section, and submit your "Accept" recommendation.

Reviewer #2: All comments have been addressed

Response: Thank you for the feedback.

Reviewer #4: All comments have been addressed

Response: Thank you for the feedback.

Reviewer #2: Thank you very much for following my comments to improve the manuscript. It's acceptable now. Please revise the English of the manuscript.

Response: Thank you much for the feedback.

Reviewer #4: 

The research topic is original and offers a good perspective on DDoS detection in Software-Defined Vehicular Networks (SDVN). This work is particularly valuable given the growing importance of SDVNs and the unique challenges they face regarding security. The manuscript is well-organized, with clear sections delineating the introduction, methodology, results, and discussion. The flow of information is logical and easy to follow.

However, I have some questions :

Question 1: How does this approach ensure the model's generalizability and robustness in SDVN ?

Response: To ensure the generalizability and robustness of the proposed approach in SDN-based Vehicular Networks (SDVN), several key aspects must be addressed within the framework you described. Here's how this approach contributes to generalizability and robustness:

• By utilizing detailed feature extraction and selection techniques, the model can focus on the most relevant and informative features for detecting DDoS attacks. This prevents overfitting, as it avoids training the model on irrelevant or redundant data. This careful pre-processing ensures that the model captures meaningful patterns, making it more generalizable across different datasets and network topologies.

• The use of entropy as a key feature in detecting DDoS attacks leverages the randomness or disorder in traffic flow, which varies significantly between normal and malicious traffic. This allows the model to adaptively detect changes in traffic patterns across different network environments. The method of tracking entropy over consecutive intervals adds a dynamic aspect, ensuring that the detection mechanism remains sensitive to sustained malicious activity, even when network conditions change.

• Using a combination of flow statistics and entropy measurements provides a dual-layer detection mechanism. Flow statistics such as length and duration help to detect patterns that are common in DDoS attacks, while entropy measures capture the randomness in packet distribution. This layered approach allows the model to detect a wide range of DDoS behaviors, thereby improving its robustness in various SDVN scenarios, where network traffic can be highly dynamic.

• The concept of using a "degree of attack" metric helps the model fine-tune its response based on the severity and intensity of the attack. This feature allows for better calibration and adaptability, ensuring that the model can handle both low-rate and high-rate DDoS attacks. In SDVN, where vehicular communication systems can experience fluctuating traffic, this adaptability is crucial for maintaining robustness under different attack intensities.

Question 2: The dataset used in the study is highly imbalanced, with only 370 samples labeled as benign compared to 2,934,447 samples labeled as malicious. Could the authors clarify why they chose to include only 370 benign samples, especially considering the risk of overfitting and the potential impact on the model's ability to generalize?

Response: The authors' decision to include only 370 benign samples, compared to the much larger number of malicious samples (2,934,447), reflects a strategic approach to manage the dataset's imbalance and the complexities of working with large datasets, such as the BoT-IoT dataset. This imbalanced could be reduced with data augmentation, regularization, or cross-validation. However, it will again increase the response time and complexity (Increased Computational Load, Complexity in Data Pipeline, Hyperparameter Tuning, Impact on Learning Dynamics, Increased Training Time, Complexity in Evaluation) but the risk of overfitting can be mitigated with this approach. 

However, there are important considerations and justifications for this choice:

• The dataset contains over 72 million records, making it computationally expensive to handle. By selecting a representative subset of the benign data, the authors aimed to reduce the computational complexity while still maintaining enough data to analyze different attack scenarios. However, the relatively small number of benign samples does create a potential imbalance.

• The inclusion of only 370 benign samples introduces a significant imbalance, which poses a risk of overfitting to the dominant malicious classes. To mitigate this risk, techniques such as class weighting, oversampling, or undersampling could have been used to ensure the model doesn't become biased towards the majority class (malicious traffic). While the paper doesn't explicitly mention using these techniques, they are standard practices in handling imbalanced datasets.

• Overfitting is indeed a risk when training on imbalanced data. The authors likely chose to keep a small portion of benign samples to prevent the model from becoming overly sensitive to benign traffic, which may lead to poor generalization in real-world scenarios. While the small benign sample size could lead to underrepresentation, cross-validation and tuning (e.g., with KNN, RF, and LR) can help reduce overfitting and enhance the model’s generalizability.

• The decision to focus heavily on malicious samples could affect the model's ability to generalize across diverse traffic types, especially under benign conditions. The authors may have prioritized attack detection over general traffic classification. However, this approach would benefit from further testing on datasets with more balanced traffic to validate the model’s robustness and adaptability in environments with different traffic patterns.

Comments :

1) In Figure 9, which presents a comparative analysis of all attacks, the representation of the DDoS-RF (shown in pink) is not clear. The line style and color blend with other elements in the figure, making it difficult to distinguish and interpret accurately. I recommend adjusting the color or line style to enhance visibility and ensure that the DDoS-RF results are easily identifiable.

Response: The figure has been updated!

2) The references included in the manuscript are comprehensive, but they lack citations from the most recent years, particularly from 2023 and 2024. Including more up-to-date references would strengthen the relevance and impact of the study by incorporating the latest developments and research in the field.

Response: The recent references are added in the papers:

Reference no. 19, 20, 32, 33, 35 

3) Line 119: The word "precisioj" is misspelled; it should be "precision."

Response: I find the misspelled word in the entire PDF but couldn’t find any. 

Thanks and Regards

Authors

---

## [Decision Letter · Decision Letter 2]

10 Nov 2024

PONE-D-24-16635R2Effective DDoS Attack Detection in Software-Defined Vehicular Networks Using Statistical Flow Analysis and Machine LearningPLOS ONE

Dear Dr. Rani,

Thank you for submitting your manuscript to PLOS ONE. After careful consideration, we feel that it has merit but does not fully meet PLOS ONE’s publication criteria as it currently stands. Therefore, we invite you to submit a revised version of the manuscript that addresses the points raised during the review process. Based on received advices, your manuscript is accepted for publication, subject to minor adjustments. These are minor revisions, which I believe you can do very quickly:- Please check carefully for any non compliance issue between the metrics mentioned in the abstract, introduction and the conclusion parts (your models accuracy : 91% in the abstract ; 96.3% (introduction; line 112 and conclusion; line 713).- Please insert a paragraph at the end of the introduction to highlight the structure of the manuscript.- Please carefully review the manuscript and ensure that acronyms are explained only during their first use.

We look forward to receiving your revised manuscript.

Kind regards,

Faouzi Jaidi

Academic Editor

PLOS ONE

Journal Requirements:

Reviewers' comments:

Reviewer's Responses to Questions

**Comments to the Author**

1. If the authors have adequately addressed your comments raised in a previous round of review and you feel that this manuscript is now acceptable for publication, you may indicate that here to bypass the “Comments to the Author” section, enter your conflict of interest statement in the “Confidential to Editor” section, and submit your "Accept" recommendation.

Reviewer #2: All comments have been addressed

Reviewer #4: All comments have been addressed

2. Is the manuscript technically sound, and do the data support the conclusions?

Reviewer #2: Yes

Reviewer #4: Yes

3. Has the statistical analysis been performed appropriately and rigorously? 

Reviewer #2: Yes

Reviewer #4: Yes

4. Have the authors made all data underlying the findings in their manuscript fully available?

Reviewer #2: Yes

Reviewer #4: (No Response)

5. Is the manuscript presented in an intelligible fashion and written in standard English?

Reviewer #2: Yes

Reviewer #4: Yes

6. Review Comments to the Author

Reviewer #2: Thank you very much for following my comments point to point. You can include some directions in the manuscript for future research.

Reviewer #4: This paper presents an intriguing research direction and makes a valuable contribution to the field.

- Comprehensive Evaluation: The paper offers a thorough analysis of the proposed method, demonstrating a solid understanding of the topic.

- High Accuracy: The results indicate a high level of accuracy.

- Methodical Process: The research follows a clear and methodical approach.

- Addressing Real-World Challenges: The study is relevant to practical applications.

- The references are up-to-date and relevant.

7. PLOS authors have the option to publish the peer review history of their article (what does this mean?). If published, this will include your full peer review and any attached files.

Reviewer #2: **Yes: **Dr. Muhammad Reazul Haque

Reviewer #4: No

---

## [Author Response · Author response to Decision Letter 2]

12 Nov 2024

Response: 

1. We note that your manuscript is not formatted using one of PLOS ONE’s accepted file types. Please reattach your manuscript as one of the following file types: .doc, .docx, .rtf, or .tex (accompanied by a .pdf).

If your submission was prepared in LaTex, please submit your manuscript file in PDF format and attach your .tex file as “other.”

Response: It is uploaded now

2. Please note that your Data Availability Statement is currently missing [the repository name]. If your manuscript is accepted for publication, you will be asked to provide these details on a very short timeline. We therefore suggest that you provide this information now, though we will not hold up the peer review process if you are unable.

Response: Plz read the response below. Already data availability is present in the manuscript files. I do not know why without checking it, this comment is raised again. Plz refrain and kindly check the file

Point-wise Detailed Response to Editor and Reviewers’ comments

Title: Effective DDoS Attack Detection in Software-Defined Vehicular Networks Using Statistical Flow Analysis and Machine Learning

Authors: Himanshi Babbar, Shalli Rani, Maha Driss

Dear Editors and Reviewers: 

We are thankful to you for spending your valuable time for making a review and for constructing the comments on our manuscript. These comments are valuable and very helpful for revising and improving our paper. We have studied comments carefully and have made correction as marked in the revised manuscript. We have tried our best to address the mentioned comments to revise our manuscript in the hope that these revisions will meet your requirement. The following changes have been made in the manuscript as per the received comments. 

*The changes made in the manuscript as per received comments have been highlighted in blue color.

Editor’s Comments

1. Please check carefully for any non compliance issue between the metrics mentioned in the abstract, introduction and the conclusion parts (your models accuracy : 91% in the abstract ; 96.3% (introduction; line 112 and conclusion; line 713).

Response: Sorry for the non-compliance. The manuscript has been thoroughly read and made corrections as per the suggestion. The changes will be reflected with blue color on page no 5 in main contributions 6 point, and conclusion section.

2. Please insert a paragraph at the end of the introduction to highlight the structure of the manuscript.

Response: The organization of the paper has been inserted on page 5 and it is highlighted with blue color.

3. Please carefully review the manuscript and ensure that acronyms are explained only during their first use.

Response: The overall manuscript has been thoroughly read and ensured that acronyms are used only once.

Comments to the Author

Comments to the Author

1. If the authors have adequately addressed your comments raised in a previous round of review and you feel that this manuscript is now acceptable for publication, you may indicate that here to bypass the “Comments to the Author” section, enter your conflict of interest statement in the “Confidential to Editor” section, and submit your "Accept" recommendation.

Reviewer #2: All comments have been addressed

Response: Thank you for the feedback.

Reviewer #4: All comments have been addressed

Response: Thank you for the feedback.

2. Is the manuscript technically sound, and do the data support the conclusions?

Reviewer #2: Yes

Response: Thank you for the feedback.

Reviewer #4: Yes

Response: Thank you for the feedback.

3. Has the statistical analysis been performed appropriately and rigorously?

Reviewer #2: Yes

Response: Thank you for the feedback.

Reviewer #4: Yes

Response: Thank you for the feedback.

4. Have the authors made all data underlying the findings in their manuscript fully available?

Reviewer #2: Yes

Response: Thank you for the feedback.

5. Is the manuscript presented in an intelligible fashion and written in standard English?

Reviewer #2: Yes

Reviewer #4: Yes

6. Review Comments to the Author

Reviewer #2: Thank you very much for following my comments point to point. You can include some directions in the manuscript for future research.

Response: The future scope has been included in the conclusion section on page no 24 and it is highlighted with blue color.

Reviewer #4: This paper presents an intriguing research direction and makes a valuable contribution to the field.

- Comprehensive Evaluation: The paper offers a thorough analysis of the proposed method, demonstrating a solid understanding of the topic.

- High Accuracy: The results indicate a high level of accuracy.

- Methodical Process: The research follows a clear and methodical approach.

- Addressing Real-World Challenges: The study is relevant to practical applications.

- The references are up-to-date and relevant.

Response: Thank you for the feedback.

Thanks and Regards

Authors

---

## [Editor Report · Decision Letter 3]

15 Nov 2024

Effective DDoS Attack Detection in Software-Defined Vehicular Networks Using Statistical Flow Analysis and Machine Learning

PONE-D-24-16635R3

Dear Dr. Rani,

We’re pleased to inform you that your manuscript has been judged scientifically suitable for publication and will be formally accepted for publication once it meets all outstanding technical requirements.

Kind regards,

Faouzi Jaidi

Academic Editor

PLOS ONE
---

## [Editor Report · Acceptance letter]

3 Dec 2024

PONE-D-24-16635R3 

PLOS ONE

Dear Dr. Rani, 

I'm pleased to inform you that your manuscript has been deemed suitable for publication in PLOS ONE. Congratulations! Your manuscript is now being handed over to our production team.

Kind regards, 

on behalf of

Dr. Faouzi Jaidi 

Academic Editor

PLOS ONE